# Quality Evaluation of the Oil of *Camellia* spp.

**DOI:** 10.3390/foods11152221

**Published:** 2022-07-26

**Authors:** Jing Yu, Heqin Yan, Yougen Wu, Yong Wang, Pengguo Xia

**Affiliations:** 1Key Laboratory for Quality Regulation of Tropical Horticultural Plants of Hainan Province, College of Horticulture, Hainan University, Haikou 570228, China; yujinghxy@163.com (J.Y.); yhqin0502_03@163.com (H.Y.); wygeng2003@163.com (Y.W.); 2Ministry of Education Key Laboratory for Ecology of Tropical Islands, College of Life Sciences, Hainan Normal University, Haikou 571158, China; 3Key Laboratory of Plant Secondary Metabolism and Regulation of Zhejiang Province, College of Life Sciences and Medicine, Zhejiang Sci-Tech University, Hangzhou 310018, China

**Keywords:** *Camellia* spp., minor compounds, fatty acid composition, antioxidant activity, quality

## Abstract

The oil of *Camellia* spp. has become a well-known high-quality edible oil because of its rich nutrition. It is of great significance to breed fine varieties of *Camellia* spp. for the sustainable growth of the *Camellia* spp. industry. This study mainly evaluated the quality and antioxidant capacity of the camellia seed from several sources. The fatty acid composition and main active components of 40 kinds of *C. oleifera*, *C. vietnamensis*, *C. osmantha*, and *C. gigantocarpa* seeds, and so on, from different regions, were tested using GC–MS and HPLC. The quality of different *Camellia* spp. germplasm resources was comprehensively evaluated using multiple indices. The unsaturated fatty acid content and the antioxidant capacity of *C. vietnamensis* from Hainan were higher than those of *C. oleifera* Abel. In addition, there were a few differences in the fatty acid compositions of *Camellia* spp. oil from different species. Correlation analysis confirmed that rutin, total saponin, total flavonoids, squalene, and vitamin E were strongly correlated to the antioxidant capacity of *Camellia* spp. In the comprehensive evaluation, the best quality and strongest antioxidant activity were found for Chengmai Dafeng (*C. vietnamensis*). These methods in the study were applied for the first time for the quality evaluation of the Camellia spp. species. This study provided new insights into the quality evaluation of the *Camellia* spp. species, thus facilitating further development of variety breeding along with quality evaluation.

## 1. Introduction

*Camellia* spp., also known as “Oriental olive oil” [1,2], is an evergreen shrub or small tree of the family Theaceae. Because of its high oil content, *Camellia* spp. is also known as one of the four woody oil plants, which also includes *Olea europaea*, *Cocos nucifera* L. and *Elaeis guineensis* Jacq. *Camellia* spp. has a long history of cultivation in its native country China, where it is mainly distributed in the Yangtze River Basin [3,4]. In addition, it is also scattered in Vietnam, Myanmar, Thailand, Malaysia, and Japan [5]. The main cultivated varieties are *C. oleifera* Abel., *C. vietnamensis* Huang, *C. gigantocarpa*, and *C. chekiangoleosa* [3]. *Camellia* spp. is a typical plant resource in China. It has high nutritional and medicinal value, and it has a long history of production and development [6].

*Camellia* spp. seeds are rich in unsaturated fatty acids, peptides, minerals, and vitamins, of which more than 80% of the constituents are unsaturated fatty acids, comprising mainly oleic acid, linoleic acid, and linolenic acid. Oleic acid, which can bate blood vessels and prevent cardiovascular and cerebrovascular diseases, accounts for more than 50% of the whole fatty acid content [7]. Furthermore, *Camellia* spp. oil contains squalene, vitamin E, tea saponin, tea polyphenols, sterol, and β-amyrin as well as other substances that are beneficial to the human body [8,9,10]. Squalene, used as a precursor to synthesize steroids, regulates human cholesterol metabolism, protects the skin, and reduces blood sugar levels [11]. Vitamin E, an umbrella term for tocopherols, tocotrienols and derivatives with physiological activities of tocopherols, has four homologs in plants (α, β, γ, and δ tocopherol) [12]. It is capable of protecting T lymphocytes and erythrocytes, inhibiting free radical oxidation, and reducing the risk of myocardial and cerebral infarction [12]. Tea saponin has antibacterial [13], anticancer [14], and antioxidation effects [15,16], and it can confer ultraviolet-radiation resistance [17]. Therefore, *Camellia* spp. can be applied in the food, cosmetics, and medical industries.

*Camellia* spp. oil is the main product of *Camellia* spp. seeds, comprising nearly 50% of the weight of dried kernel [18]. At present, *Camellia* spp. oil is mainly used for the production of edible oil [19]. The main component of edible oil is fatty acids, and the composition of fatty oils is the most important determinant affecting the quality of plant edible oil. *Camellia* spp. is a unique woody oil plant in China and has high oil content and good quality. Ma et al. isolated 67.7–76.7% oleic acid, 82–84% unsaturated fatty acids, 68–77% monounsaturated fatty acids, and 7–14% polyunsaturated acids from *C. oleifera* oil [2]. The total unsaturated fatty acids content is significantly higher than that of the saturated fatty acids. In addition to the fatty acid composition, the type and content of other bioactive ingredients contained in the oil are also important indicators of the comprehensive quality of the oil.

At present, researchers have studied the following aspects of *Camellia* spp.: its fatty acid composition [2,20], the function of its active ingredients, and its antioxidant capacity [21,22]. However, there are few studies evaluating the quality and antioxidant capacity of *Camellia* spp. from different regions. Therefore, in this study, the active components of *Camellia* spp. from different areas were extracted and analyzed; their fatty acid composition, bioactive components, DPPH radical scavenging ability, ABTS radical scavenging ability, and ferric reducing antioxidant power were compared. Finally, the quality of different *Camellia* spp. germplasm resources were comprehensively evaluated through multiple indexes, thus providing a theoretical basis for quality improvement as well as the development and application of *Camellia* spp. components.

## 2. Materials and Methods

### 2.1. Materials

A total of 40 *Camellia* spp. samples in seed maturation were provided by Prof. Kaibing Zhou in 2018 for this study, including 26 *C. vietnamensis* Huang varieties, 7 *C. oleifera* Abel., 2 *C. osmantha*, 1 *C. gigantocarpa* Hu et Huang, 1 *C. chekiangoleosa*, 1 *C. gauchowensis*, 1 *C. fangchengensis* S.Y. Liang et Y.C. Zheng, and 1 *C. sinensis* (L.) O. Ktze. Sample numbers are shown in Table 1.

Squalene, vitamin E, stigmasterol, β-amyrin, 3,4-dihydroxyphenylacetic acid, epicatechin, tea saponin, rutin, quercetin, camellianin A standards, 1,1-diphenyl-2-picrylhydrazyl (DPPH), 2,2-azino-bis (3-ethylbenzothiazoline-6-sulfonic acid) diammonium salt (ABTS), and Trolox were purchased from Sigma–Aldrich Chemical Co. Ltd. (Shanghai, China). Other reagents and solvents were obtained from Shanghai Macklin Biochemical Co. Ltd. (Shanghai, China).

### 2.2. Treatment of Camellia spp. Seeds

*Camellia* spp. seeds were dried in the oven at 50 °C, then shelled and crushed. The processed seeds were loaded into the filter paper package (Soxhlet extractor) with petroleum ether (boiling point: 60–90 °C) for extraction at 50 °C for 5 h to obtain *Camellia* spp. oil and to obtain a deoiled powder [23,24,25].

### 2.3. Fatty Acid Composition and Content Analysis

Fatty acid methyl esterification of *Camellia* spp. oil was carried out with reference to GB/T 17376-2008. Briefly, 40 µL of an oil sample was put into a 5 mL centrifuge tube, to which 2 mL of 1 mol·L^−1^ KOH/CH3OH solution and 2 mL of 5% sulfuric acid solution in a cold bath were added. The centrifuge tube was inverted, and the mixture was incubated in a water bath set to a constant temperature of 55 °C for 1 h. The tube was manually shaken every 10 min for 5 s and cooled in a cold bath. Thereafter, 2 mL of n-hexane was added to the mixture, which was then vortexed for 5 min. The sample was centrifuged for 5 min at 994× *g* [26]. The supernatant was filtered through a membrane solution filter (0.22 μm) and subjected to GC analysis.

The mixture was analyzed using an Agilent 7890B-7000B gas chromatography machine equipped with an Agilent122-1032G column (30 m × 0.25 mm × 0.25 μm) under the following temperature conditions: 100 °C for 1 min, followed by ramping of 6 °C·min^−1^ to 240 °C, and then maintenance at 240 °C for 12 min. The detector temperature was set to 270 °C. The flow rate of air was 450 mL·min^−1^, the flow rate of hydrogen was 40 mL·min^−1^, and the flow rate of the tail blow was 45 mL·min^−1^. The injection volume was 1.0 μL. The levels of fatty acids are reported as relative proportions.

### 2.4. Determination of the Components and Content of Camellia spp.

#### 2.4.1. Determination of Bioactive Compounds Using GC–MS and HPLC

*Camellia* spp. oil (0.1 g) was diluted with *n*-*hexane* to 5 mL and filtered through a membrane solution filter (0.22 μm) prior to GC–MS analysis. The GC–MS conditions are based on the method of Ye [27]. The mixture was analyzed using an Agilent 7890B-7000B gas chromatograph equipped with an HP-5MS column (30 m × 0.25 mm × 0.25 μm) under the following temperature conditions: 60 °C for 1 min, followed by ramping of 6 °C·min^−1^ to 270 °C, and then maintenance at 270 °C for 2 min. The transfer line and ion resource temperatures were set to 250 °C. The injected volume was 1 μL, splitless. The flow rate of pure helium (99.99%), the carrier gas, was 1.00 mL·min^−1^. The mass spectrometry conditions were as follows. The ion source was EI and the temperature was 230 °C. The quadrupole temperature was 150 °C, ionization voltage was 70 eV, and the emission current was 34.60 μA. The multiplier voltage was 2000 V. Data were obtained continuously in the full-scan mode in the mass range of 50–450 (m/z). On the HPMSD chemical workstation, compounds were tentatively identified using the NIST2005 MS and WILEY275 MS libraries, and their relative contents were calculated by normalization of chromatographic peak area [28].

*Camellia* spp. de-oiled powder (0.1 g) was loaded into a 5 mL centrifuge tube containing 1.5 mL of 50% methanol solution, ultrasonically extracted for 30 min, and centrifuged at 2030× *g* for 5 min at 20 °C. The supernatant was removed and loaded into an elastic-quartz capillary column and then filtered through the injection filter. The HPLC conditions were as follows: the separation column was ODS C18 (0.46 mm × 150 mm), and the detection wavelength was 280 nm. A dual pump system was used for mobile phases A and B. Mobile phase A was 0.2% ice/acetic acid, and mobile phase B was acetonitrile. The gradient program was as follows: 0–10 min, isocratic 10% B; 10–18 min, isocratic 20% B; 18–28 min, isocratic 35% B; 36–42 min, isocratic 65% B; 42–49 min, isocratic 100% B; 49–56 min, isocratic 10% B. The chromatography column temperature was 35 °C, and the flow rate was 1.0 mL·min^−1^.

#### 2.4.2. Determination of Total Phenolic and Total Flavonoid Content

The 0.3 g *Camellia* spp. degreasing powder was extracted by 4.5 mL 50% methanol extraction solution, then treated by ultrasonic wave (water bath 60 °C, power 100 W, frequency 40 kHz) for 30 min, filter, repeat the above steps for 3 times, combine the filtrate, vacuum concentrate at 45 °C, add equal volume of extraction solution, centrifuge at 1697× *g* for 15 min, suck the supernatant, and store at 20 °C for use [27].

The total phenol content in *Camellia* spp. was detected using the Folin–phenol method with gallic acid as the standard. The method of Ye [27] was used to prepare the standard curve and sample liquid as well as to determine the total phenol content.

Rutin was used as the standard material to detect the total flavonoid content, and the method of Ye [27] was used to prepare the standard curve and to determine the total flavonoid content.

### 2.5. Determination of Antioxidant Activity

The 1 g *Camellia* spp. degreasing powder was extracted by 15 mL 50% methanol extraction solution, then treated by ultrasonic wave (water bath 60 °C, power 100 W, frequency 40 kHz) for 30 min, filter, repeat the above steps for 3 times, combine the filtrate, vacuum concentrate at 45 °C, add equal volume of extraction solution, centrifuge at 1697× *g* for 15 min, suck the supernatant, and store at 20 °C for use [27].

#### 2.5.1. DPPH (1, 1-diphenyl-2-picrylhydrazyl) Radical Scavenging Assay

The antioxidant capacity was determined using 1,1-diphenyl-2-picrylhydrazyl (DPPH) following the experimental method of Zantar et al. [29]. The diluted Trolox standard (100 μL) and sample solution were added to 3 mL of DPPH solution, and the same volume of 50% methanol was used as the blank control. The absorbance values were measured at 1 h at the wavelength of 517 nm. The average absorbance values of triplicate measurements were used for data analysis. The results were converted into the Trolox equivalent antioxidant capacity (unit of mmol·L^−1^ Trolox·g^−1^ DW), and the standard curve was plotted.

#### 2.5.2. ABTS (3-ethylbenzothiazoline-6-sulfonicacid) Radical Scavenging Assay

The antioxidant capacity was determined using 2,20-azinobis-(3-ethylbenzothiazoline-6-sulfonicacid) (ABTS) following the experimental method of Ye [27]. The diluted Trolox standard (150 μL) and sample solution (50 μL) were added to 3 mL of the ABTS working solution, and the same volume of 50% methanol was used as the blank control. The mixtures were left in the dark for 1 h, and the absorbance values were measured at 734 nm. The average absorbance values of triplicate measurements were used for data analysis. The results were converted into the Trolox equivalent antioxidant capacity (unit of mmol·L^−1^ Trolox·g^−1^ DW), and the standard curve was plotted.

#### 2.5.3. Ferric Reducing Antioxidant Power (FRAP) Assay

The antioxidant capacity in terms of the ferric reducing antioxidant power (FRAP) of *Camellia* spp. extracts was determined following the method described by Stojanovic et al. [30]. The diluted FeSO_4_ standard solution (2 mL) and sample solution (40 μL) were mixed with 3 mL of the FRAP working solution, and the same volume of 50% methanol was used as the blank control. The mixtures were left in the dark for 1 h, and the absorbance values were measured at 593 nm. The average absorbance values of triplicate measurements were used for data analysis. The results were converted into the Trolox equivalent antioxidant capacity (unit of mmol·L^−1^ FeSO_4_·g^−1^ DW), and the standard curve was drawn.

### 2.6. Data Statistical Analysis

All analyses were conducted in triplicate, with results expressed as average values ± standard deviation (AVG ± SD). Correlation analysis and the main component of *Camellia* spp. were analyzed using the SPSS software. *Camellia* spp. from Hainan Island and inland China were evaluated by assessing the quality and antioxidant capacity.

## 3. Results

### 3.1. The Comparative Analysis of the Oil Content

As shown in Table 2, the study found that most of the oil content (*w*/*w*) of the samples was concentrated between 40% and 55%, and the average oil content was 46.87%. CMDF (*C. vietnamensis*) had the highest oil content (58.96%), and BT (*C. sinensis*) had the lowest oil content (30.22%). The oil content of the seven *C. oleifera* Abel samples was between 40% and 45%, while 10 *C. vietnamensis* Huang had more than 50%, namely QZ1 (*C. vietnamensis*), QZ4 (*C. vietnamensis*), QZ8 (*C. vietnamensis*), NC1 (*C. vietnamensis*), ND3 (*C. vietnamensis*), HS1 (*C. vietnamensis*), HS2 (*C. vietnamensis*), FS2 (*C. vietnamensis*), CMDF (*C. vietnamensis*), and BWL (*C. vietnamensis*). It showed that this *C. vietnamensis* had good characteristics in terms of oil content. In addition, the oil content of the three samples of ND3 (*C. vietnamensis*), FS2 (*C. vietnamensis*), and CMDF (*C. vietnamensis*) are higher than 55%, which can be used for further research and analysis. There are great differences in the oil content of dried *Camellia* spp. seeds in different regions. Each region should choose the varieties suitable for the region according to their own conditions, such as soil, temperature, cultivation methods, etc., to breed good varieties.

### 3.2. Fatty Acid Composition

#### 3.2.1. Standard Samples

The fatty acid composition of the standard samples was determined using a 7890N gas chromatograph. As shown in Appendix A, the fatty acid component was based on the retention time of the peak and their relative contents were calculated by normalization of the chromatographic peak area.

#### 3.2.2. *Camellia* spp. Oil Samples

In all studied samples, the main fatty acids identified were oleic acid, linoleic acid, palmitic acid, and stearic acid. These four fatty acids accounted for more than 98% of the fatty acid content. Oleic acid was the primary unsaturated fatty acid of *Camellia* spp. oil, accounting for 86.23% (QZ2, *C. vietnamensis*) to 1.23% (BT, *C. sinensis*) of the total fatty acid content. By analyzing Table 3, we determined that the unsaturated fatty acids (UFA) contained in *Camellia* spp. seed oil included oleic acid, linoleic acid, linolenic acid, and palmitoleic acid, and the saturated fatty acids included myristic acid, arachidic acid, palmitic acid, and stearic acid.

#### 3.2.3. Correlation between the Four Major Fatty Acids

Pearson correlation analysis was performed on four fatty acids constituting more than 98% of the total fatty acid content using the SPSS software, and the result is shown in Table 4. As shown in Table 4, the oleic acid content had a significant negative correlation with the linoleic acid and palmitic acid contents. However, the linoleic acid content had a significant positive correlation with the palmitic acid content and a significant negative correlation with the stearic acid content. It was concluded that oleic acid content in camellia oil could affect linoleic acid and palmitic acid contents in camellia oil, and linoleic acid content could affect stearic acid and palmitic acid contents.

### 3.3. Analysis of the Bioactive Components

#### 3.3.1. The Minor Compounds from the Oil Using GC–MS

The minor compounds of the *Camellia* spp. samples were analyzed using GC–MS. The relative contents of minor compounds (squalene, α-tocopherol, β-sitosterol, and β-amyrin) were calculated by normalization of the chromatographic peak area. We focused on two active components, squalene and α-tocopherol, because they both have good medicinal values. As shown in Figure 1A, the squalene content varied greatly among the different samples. The five samples with high squalene contents were CMDF (*C. vietnamensis*), QZ1 (*C. vietnamensis*), HS1 (*C. vietnamensis*), NC2 (*C. vietnamensis*), and QZ8 (*C. vietnamensis*). The difference in relative contents of β-sitosterol between the 40 samples was not as great as that of squalene, and those with relative contents greater than 6% were BB (*C. gigantocarpa*), GZ (*C. gauchowensis*), HN (*C. oleifera*), and QZ4 (*C. vietnamensis*). The five samples with high α-tocopherol contents were QH (*C. vietnamensis*), HS3 (*C. vietnamensis*), QZ9 (*C. vietnamensis*), CMDF (*C. vietnamensis*), and QZ1 (*C. vietnamensis*). The above (CMDF, QZ1, HS1, NC2, QZ8, QH, HS3, QZ9) can be used as suitable samples for further study.

#### 3.3.2. The Polar Compounds from the Extraction Meal Determined Using HPLC

The peak areas of HPLC chromatograms were used to determine the contents of the active components in the standards, as shown in Appendix A. The total saponin content included tea saponins A and B. The standard curve for each standard was made according to the peak area and known concentration of the standard, as shown in Appendix A.

Concentrations (C; mg·L^−1^) of the active components were obtained through peak area normalization, and the content of each active component was calculated from these concentrations, as shown in Figure 1B. Content (%) = (C × 1.5 mL/100 mg) × 100%.

As shown in Figure 1B, the content of tea saponin in the test substance was the highest, and the highest average value was 208.50 mg·g^−1^. QZ3 (*C. vietnamensis*) had the highest tea saponin content (374.86 mg·g^−1^), and FS2 (*C. vietnamensis*) had the lowest tea saponin content. There were common peaks in the chromatograms of 40 samples, although slightly different peaks were observed for some special samples. For example, the HPLC chromatogram of BB (*C*. *gigantocarpa*) differed greatly from those of the other samples, and the target active component was not detected for BB (*C*. *gigantocarpa*). Therefore, BB (*C*. *gigantocarpa*) was considered to belong to a special species, and its specific composition was unknown.

#### 3.3.3. Total Phenol and Flavonoid Contents

The contents of total phenols and flavonoids in 40 *Camellia* spp. samples were determined using a 752N UV spectrophotometer. The linear fit of the standard curve using gallic acid as the standard was y = 7.7093x (R2 = 0.9983), while the linear fit of the standard curve using rutin as the standard was y = 11.451x (R2 = 0.9946). The measured absorption values and the standard curve were used to calculate the total phenol and total flavonoid concentrations (g·L^−1^) in the sample, which were used to calculate the total phenol and total flavonoid contents, as shown in Figure 2A.

As shown in Figure 2A, among the 40 kinds of *Camellia* spp. tested, the total phenol content was between 6.23% and 10.87%, and the average total phenol content was 8.51%. In particular, the highest total phenol content (10.87%) was found for CR3 (*C. oleifera*), and the lowest (6.23%) for GZTR (*C. oleifera*). The total was between 2.84% and 8.68%, and the average was 5.12%. The highest flavonoid content (8.68%) was found for CMDF (*C. vietnamensis*), and the lowest (2.84%) for XY (*C. oleifera*).

### 3.4. Antioxidant Capacity

The free radical scavenging capacities of *Camellia* spp. seeds were determined using three methods (DPPH, ABTS, and FRAP). The standard linear equations are shown in Appendix A. The measured light absorption values and the linear equations were used to obtain the antioxidant values of different varieties of *Camellia* spp. The results are shown in Figure 2B.

As shown in Figure 2B, the range of antioxidant capacities from the DPPH test was relatively large, between 0.20 and 1.53 mmol·L^−1^ Trolox·g^−^^1^ DW, with an average of 1.35 mmol·L^−1^ Trolox·g^−1^ DW. The highest antioxidant capacities were found for CMDF (*C. vietnamensis*), FS2 (*C. vietnamensis*), BT (*C. sinensis*), and GZ (*C. gauchowensis*), and the lowest for XY (*C. oleifera*). From the ABTS assay, the antioxidant values were between 0.27 and 1.64 mmol·L^−1^ Trolox·g^−1^ DW, with an average of 1.30 mmol·L^−1^ Trolox·g^−1^ DW. QZ1 (*C. vietnamensis*) had the highest antioxidant capacity, followed by BT (*C. sinensis*), and BB (*C. gigantocarpa*) had the lowest antioxidant capacity. Overall, obvious differences were not observed in the ABTS clearance ability among the varieties from different areas. Using the FRAP method, the antioxidant value of *Camellia* spp. varieties from 40 different producing areas was between 1.41 and 3.90 mmol·L^−1^ Trolox·g^−1^ DW. CMDF (*C. vietnamensis*) had the highest value, followed by GZ (*C. gauchowensis*), which was relatively high, and XY (*C. oleifera*) had the lowest value. According to these measurements, more pronounced differences in the antioxidant capacity among different cultivars were observed with the FRAP method than with the ABTS and DPPH methods.

### 3.5. Correlations between Bioactive Components and Free Radical Scavenging Capacity

The correlation between active components and antioxidant capacity was analyzed using SPSS software (26.0, SPSS Inc., MD-NC119 Armonk, NY, USA). As shown in Figure 2C, rutin had a significant negative correlation with DPPH clearance, ABTS clearance, and FRAP reduction. There was a significant positive correlation between total tea saponin and DPPH clearance, ABTS clearance, and FRAP reduction. Squalene and α-tocopherol showed a significant positive correlation with FRAP reduction. DPPH clearance, ABTS clearance, and FRAP reduction all showed a significant positive correlation with each other. Among them, the total flavonoids were highly correlated with DPPH and ABTS clearances. Therefore, the results obtained using the DPPH and ABTS methods indicated that the antioxidant capacity of *Camellia* spp. was dependent on the contents of active substances, such as rutin, tea saponin, and total flavonoids in *Camellia* spp. It was found that the DPPH clearance capacity, ABTS clearance capacity, and Fe^3+^ reduction capacity all had significant positive correlations with the content of total flavonoids. For example, the total flavonoid content in XY (*C. oleifera*) was the lowest among the 40 *Camellia* spp. samples, and the corresponding DPPH and ABTS clearance abilities and Fe^3+^ reduction ability were relatively weak, indicating low antioxidant activity.

### 3.6. Cluster Analysis

Data on the fatty acid content of each sample were standardized and used for cluster analysis and PCA analysis (Figure 3A,B). A clear clustering tendency was observed for *Camellia* spp. samples containing similar profiles. *C. vietnamensis* and *C. oleifera* Abel. were clustered together, and thus regional divisions were more obvious. There were exceptions, for example, JX (*C. oleifera*), GX (*C. oleifera*), and GZTR (*C. oleifera*) were clustered with *C. vietnamensis*, which likely originated in mis-sampling. K13 (*C. vietnamensis*) was clustered with *C. oleifera* Abel., indicating that environmental conditions in mainland China were different from Hainan, thus affecting the fatty acid content. Data on the contents of bioactive components for each sample were standardized and used for cluster analysis and PCA analysis (Figure 3C,D). The clustering results were similar to those of fatty acids.

### 3.7. Principal Component Analysis

The 15 component indicators of the 40 *Camellia* spp. samples were standardized for principal component analysis. As shown in Appendix A, the characteristic value of the first principal component was 4.852, the variance was 32.345%, and the cumulative variance was 32.345%. The most important principal component was the first principal component, and the importance decreased from the second to the fifth principal component. The cumulative variance contribution rate of the first five principal components reached 77.262%, and thus the five main components reflected most of the information for all 15 components of *Camellia* spp. Therefore, these five principal components were selected as the comprehensive evaluation index.

The principal component load matrix reflected the extent to which each quality indicator impacted this principal component, as shown in Appendix A. Using the 0.5 principle, the first main component included α-tocopherol, total tea saponins, rutin, Camellianin A, total flavonoids, ABTS, DPPH, and FRAP. The second major component included squalene, α-tocopherol, β-sitosterol, and β-amyrin. The third major component included total phenols and total flavonoids. The fourth major component included quercetin. The fifth main component included α-tocopherol, total tea saponins, rutin, Camellianin A, total flavonoids, ABTS, DPPH, and FRAP.

The function expressions of five main components were calculated from the initial factor load matrix and the characteristic value of each main component. Using the function expression of each main component, the score values and rankings of the main components of the 40 *Camellia* spp. samples were calculated, and then the comprehensive score and comprehensive ranking of the nutritional quality of the 40 *Camellia* spp. samples were calculated from the main component comprehensive score model (Table 5). As shown in Table 5, among the 40 kinds of *Camellia* spp., the three samples with high comprehensive scores were CMDF (*C. vietnamensis*), GZ (*C. gauchowensis*), and QZ1 (*C. vietnamensis*), and the samples with low rankings were WH (*C. chekiangoleosa*) and XY (*C. oleifera*).

## 4. Discussion

According to the GC analysis results, four of the five samples with high unsaturated fatty acids contents were *C. vietnamensis*, and thus we considered that the unsaturated fatty acid content of *C. vietnamensis* was higher than that of *C. oleifera* Abel. The unsaturated fatty acid content of QZJ (*C. vietnamensis*) was higher than 90%, and the squalene and α-tocopherol contents of CMDF (*C. vietnamensis*) and QZ1 (*C. vietnamensis*) were high. Therefore, these samples are useful for further studies. Docosa-13-enoic acid is a kind of super long chain fatty acid [31]. The studies showed that docosa-13-enoic acid may affect the digestion of rapeseed oil in humans, cause myocardial damage, make the cholesterol level of adrenal tissue rise and cause fat accumulation in the heart tissue [32]. Long-term consumption of rapeseed oil with high levels of docosa-13-enoic acid can increase the risk of cardiovascular diseases [33,34]. In summary, excluding docosa-13-enoic acid, which is damaging to the heart muscle, *Camellia* seed oil is considered highly nutritious because its unsaturated fatty acid content reaches 88%, a value far higher than those of vegetable oil, peanut oil, and bean oil. This study found that the fatty acid composition and content in varieties from different regions were different, which is consistent with previous results [20,35,36]. This is likely related to the effect of different environmental conditions, especially climatic conditions, such as temperature and humidity, on the composition of fatty acids [37,38,39]. In addition, other factors, such as harvest maturity and processing methods, can affect the composition of fatty acids [40]. This study found that the composition and content of fatty acids in samples from Hainan were significantly different from those in the samples of other producing regions, which is likely attributable to the unique geographical environment, including the climate, of Hainan [6,36].

The human body contains a large number of free radicals, including reactive oxygen species, of which excessive amounts are implicated in cardiovascular and cerebrovascular diseases, diabetes, tumors, and other diseases. *Camellia* spp. contains active substances that can remove free radicals, which can delay aging to a certain extent and protect the skin from the adverse effects of ultraviolet radiation. Therefore, it is a good, natural raw material for makeup and skincare products. The antioxidant activities among these 40 *Camellia* spp. species differed greatly, even for *Camellia* spp. cultivated in the same region. Therefore, differences in cultivation and management, including the use of light, water, and fertilizer, also have significant effects on the antioxidant activity of the oil. In this study, CMDF (*C. vietnamensis*) had the strongest DPPH clearance ability, while QZ1 (*C. vietnamensis*) had the strongest ABTS clearance ability, and CMDF (*C. vietnamensis*) had the strongest Fe^3+^ reduction ability. Through comprehensive SPSS and PCA analyses, CMDF (*C. vietnamensis*) was found to have the strongest antioxidant capacity. The clustering results indicated that *C. gauchowensis* and *C. vietnamensis* clustered together, consistent with the results of Qi et al. [41], who found that the leaves, flowers, fruits, and seeds of *C. vietnamensis* from Hainan are similar to those of *C. gauchowensis*, which is mainly distributed in Guangdong and Hainan and considered a native species. This specie is important because it is the most widely distributed variety of *Camellia* spp., extending into the southern regions [41]. Therefore, we considered that the genetic relationship between *C. gauchowensis* and *C. vietnamensis* is very close.

The antioxidant capacity of *C. vietnamensis* from Hainan was found to be higher than that of *C. oleifera* Abel. Therefore, it is more beneficial to produce *Camellia* spp. in areas with high temperatures and sufficient light, which are the climatic conditions in Hainan. Moreover, the accumulation of secondary metabolites and the presence of other substances with antioxidant activity, such as flavonoids, in *C. vietnamensis* from Hainan may lead to an antioxidant activity that is higher than that of *C. oleifera* Abel. Additionally, although K13 belongs to *C. vietnamensis*, its antioxidant activity was relatively low. The inland climatic conditions under which K13 (*C. vietnamensis*) grows may result in the large difference in antioxidant activity between K13 and *C. vietnamensis* from Hainan. Differences in the antioxidant activity of *Camellia* spp. varieties from Hainan were not obvious, but may cause the small difference between total phenols and total flavonoids, although the contents of other active substances may also differ; further analysis using high-performance liquid-phase methods may be required.

## 5. Conclusions

The present study showed that the content of unsaturated fatty acids and the antioxidant capacity of *C. vietnamensis* from Hainan were generally higher than those of *C. oleifera* Abel. Additionally, this study also found that there were significant differences in the fatty acid compositions of *Camellia* spp. oil from different species. CMDF (*C. vietnamensis*) had the best quality in the comprehensive evaluation, and its antioxidant capacity was the strongest. Correlation analysis confirmed that rutin, total saponin, total flavonoids, squalene, and α-tocopherol were strongly correlated to the antioxidant capacity of *Camellia* spp. *Camellia* spp. oil is a high-quality vegetable oil and has broad application prospects. However, basic research and application of *Camellia* spp. are not comprehensive and systematic, and more functional properties of *Camellia* spp. have yet to be studied and developed. This study provides a theoretical basis for the breeding of improved varieties of *Camellia* spp. and the development and application of functional components.

## Figures and Tables

**Figure 1 foods-11-02221-f001:**
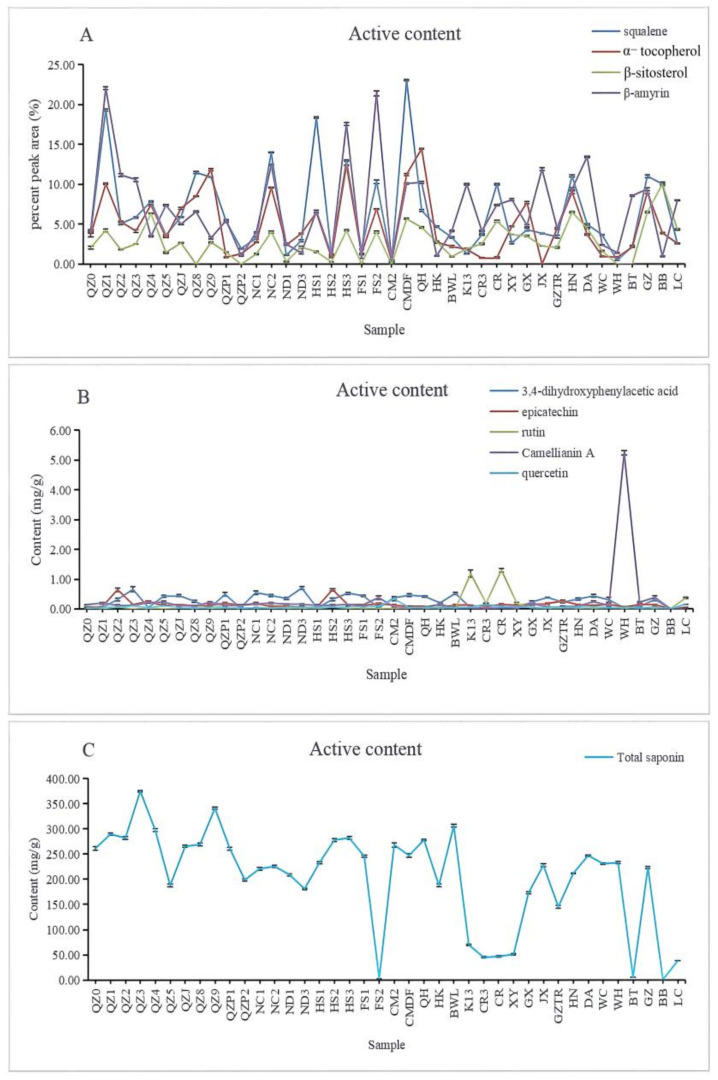
The variation tendency of active content in *Camellia* spp. from 40 samples, respectively (*p* ≤ 0.01). (**A**): The content of minor compounds from the oil by GC–MS; (**B****,C**): The content of polar compounds from the extraction meal determined by HPLC.

**Figure 2 foods-11-02221-f002:**
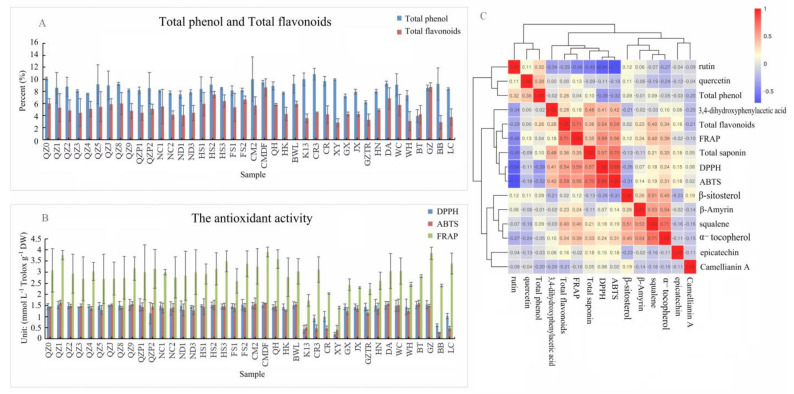
The content of total phenolic and total flavonoids and the antioxidant activities of 40 kinds of *Camellia* spp. (**A**): The content of total phenolic and total flavonoids; (**B**): The antioxidant activities (*p* ≤ 0.01); (**C**): Correlation analysis on the content of active components and antioxidation index including DPPH, ABTS, and FRAP in *Camellia* spp. samples.

**Figure 3 foods-11-02221-f003:**
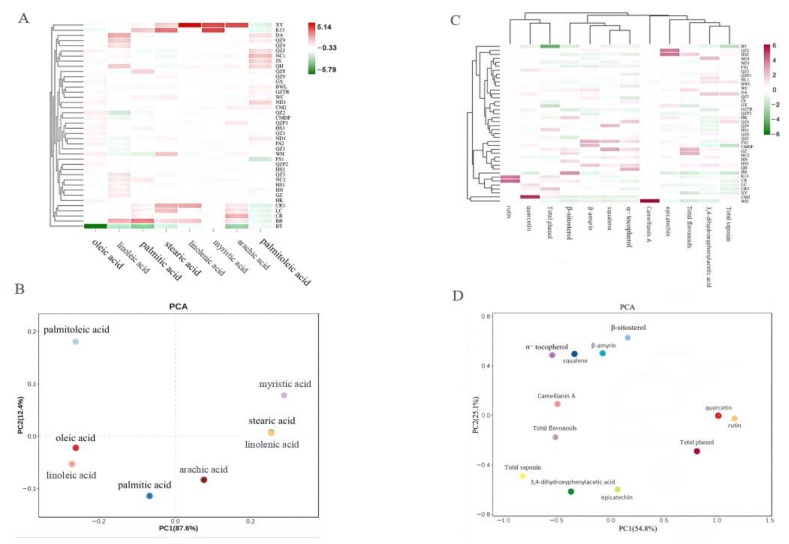
(**A**): The heatmap of fatty acids in *Camellia* spp., respectively; (**B**): The PCA analysis of fatty acids in *Camellia* spp., respectively; (**C**): The heatmap of active content in *Camellia* spp., respectively; (**D**): The PCA analysis of active content in *Camellia* spp., respectively.

**Table 1 foods-11-02221-t001:** Collection of Sample Information.

No.	Variety Name	Species Name	Region	Latitude	Longitude	Altitude (m)
QZ0	Qiongzhong 0	*Camellia vietnamensis* Huang	Wanling Town, Qiongzhong County, Hainan Province	19°08′35″ N	109°53′48″ E	183–264
QZ1	Qiongzhong 1	*Camellia vietnamensis* Huang				
QZ2	Qiongzhong 2	*Camellia vietnamensis* Huang				
QZ3	Qiongzhong 3	*Camellia vietnamensis* Huang				
QZ4	Qiongzhong 4	*Camellia vietnamensis* Huang				
QZ5	Qiongzhong 5	*Camellia vietnamensis* Huang				
QZJ	Qiongzhong J	*Camellia vietnamensis* Huang				
QZ8	Qiongzhong 8	*Camellia vietnamensis* Huang				
QZ9	Qiongzhong 9	*Camellia vietnamensis* Huang				
QZP1	Pozhai 1	*Camellia vietnamensis* Huang	Pozhai Village, Wanling Town, Qiongzhong County, Hainan Province	19°12′18″ N	109°55′31″ E	250–264
QZP2	Pozhai 2	*Camellia vietnamensis* Huang				
NC1	Nongchang 1	*Camellia vietnamensis* Huang	Changhao Farm, Wuzhishan City, Hainan Province	18°45′19″ N	109°29′24″ E	350
NC2	Nongchang 2	*Camellia vietnamensis* Huang				
ND1	Nanding 1	*Camellia vietnamensis* Huang	Nanding Village, Wuzhishan City, Hainan Province	18°49′48″ N	109°34′56″ E	556
ND3	Nanding 3	*Camellia vietnamensis* Huang				
HS1	Hongshan 1	*Camellia vietnamensis* Huang	Hongshan Village, Wuzhishan City, Hainan Province	18°51′35″ N	109°30′56″ E	620
HS2	Hongshan 2	*Camellia vietnamensis* Huang				
HS3	Hongshan 3	*Camellia vietnamensis* Huang				
FS1	Fansai 1	*Camellia vietnamensis* Huang	Fansai Village, Wuzhishan City, Hainan Province	18°50′37″ N	109°32′24″ E	556–600
FS2	Fansai 2	*Camellia vietnamensis* Huang				
CM2	Chengmai 2	*Camellia vietnamensis* Huang	Chengmai County, Hainan Province	19°44′22″ N	110°00′32″ E	0
CMDF	Chengmai Dafeng	*Camellia vietnamensis* Huang	Dafeng Town, Chengmai County, Hainan Province	19°51′12″ N	110°02′51″ E	33–48
QH	Qionghai Yangjiang	*Camellia vietnamensis* Huang	Yangjiang Town, Qionghai City, Hainan Province	19°05′59″ N	110°20′56″ E	21
HK	Haikou Dongshan	*Camellia vietnamensis* Huang	Dongshan Town, Haikou City, Hainan Province	19°44′51″ N	110°14′15″ E	30–42
BWL	Hainan Bawangling	*Camellia vietnamensis* Huang	Bawangling, Changjiang County, Hainan Province	19°13′24″ N	109°02′32″ E	90
K13	Ke 13	*Camellia vietnamensis* Huang	Guangxi Academy of Forestry (Nanning City)	22°55′10″ N	108°21′10″ E	118
CR3	Cenruan 3	*Camellia oleifera* Abel.				
CR	Cenruan Jiaxi	*Camellia oleifera* Abel.				
XY	Xinyang	*Camellia oleifera* Abel.	Xinyang City, Henan Province	32°08′54″ N	114°05′28″ E	67–105
GX	Guangxi Majiang	*Camellia oleifera* Abel.	Majiang Town, Zhaoping County, Guangxi Province	23°52′41″ N	111°02′51″ E	54
JX	Jiangxi	*Camellia oleifera* Abel.	Dongxiang County, Fuzhou City, Jiangxi Province	28°14′53″ N	116°36′12″ E	58
GZTR	Guizhou Tongren	*Camellia oleifera* Abel.	Benzhuang Town, Shiqian County, Tongren City, Guizhou Province	27°32′25″ N	107°55′41″ E	508
HN	Hunan	*Camellia oleifera* Abel.	Shaoyang City, Hunan Province	27°14′22″ N	111°28′05″ E	210–291
DA	Ding’an Xianghua Youcha	*Camellia osmantha*	Ding’an County, Hainan Province	19°40′52″ N	110°21′33″ E	42–95
WC	Wenchang Xianghua Youcha	*Camellia osmantha*	Qinglan District, Wenchang City, Hainan Province	19°32′51″ N	110°48′02″ E	9
WH	Wuzhishan Honghua Youcha	*Camellia chekiangoleosa*	Wuzhishan City, Hainan Province	18°46′29″ N	109°30′57″ E	556
BT	Baoting Chaxi Gucha	*Camellia sinensis* (L.) O. Ktze.	Tea Creek Valley, Baoting County, Hainan Province	18°38′26″ N	109°42′01″ E	42–82
GZ	Gaozhou Youcha	*Camellia gauchowensis*	Gaozhou City, Guangdong Province	21°55′08″ N	110°51′13″ E	30–66
BB	Bobai Daguo	*Camellia gigantocarpa* Hu et Huang	Bobai County, Yulin City, Guangxi Province	22°16′24″ N	109°58′33″ E	69–89
LC	Luchuan	*Camellia fangchengensis* S.Y. Liang et Y.C. Zheng	Luchuan County, Yulin City, Guangxi Province	22°19′17″ N	110°15′51″ E	101

**Table 2 foods-11-02221-t002:** The oil content of the different samples.

Sample Name	Oil (*w*/*w*: %)	Sample Name	Oil (*w*/*w*: %)	Sample Name	Oil (*w*/*w*: %)	Sample Name	Oil (*w*/*w*: %)
QZ0	48.19 ± 0.96 ^ghi^	QZP2	45.92 ± 0.31 ^ijkl^	CM2	47.64 ± 0.69 ^ghij^	JX	44.14 ± 0.52 ^lmn^
QZ1	52.81 ± 0.82 ^cd^	NC1	54.46 ± 1.42 ^bc^	CMDF	58.96 ± 0.58 ^a^	GZTR	44.01 ± 0.17 ^lmn^
QZ2	45.21 ± 0.65 ^klm^	NC2	46.82 ± 2.15 ^hijk^	QH	45.46 ± 1.09 ^jkl^	HN	40.81 ± 1.37 ^opq^
QZ3	45.12 ± 0.89 ^klm^	ND1	49.78 ± 2.54 ^fg^	HK	43.98 ± 1.31 ^lmn^	DA	46.08 ± 0.71 ^ijkl^
QZ4	51.72 ± 1.66 ^def^	ND3	58.77 ± 1.08 ^a^	BWL	51.52 ± 0.56 ^def^	WC	48.11 ± 0.43 ^ghi^
QZ5	41.35 ± 0.37 ^opq^	HS1	53.74 ± 0.68 ^bcd^	K13	40.18 ± 0.61 ^pq^	WH	41.2 ± 0.33 ^opq^
QZJ	47.82 ± 1.29 ^ghi^	HS2	53 ± 0.72 ^cd^	CR3	44.56 ± 1.29 ^klmn^	BT	30.22 ± 1.16 ^r^
QZ8	52.1 ± 0.24 ^de^	HS3	49.88 ± 0.82 ^efg^	CR	42.91 ± 0.58 ^mno^	GZ	45.04 ± 0.67 ^klm^
QZ9	40.01 ± 0.53 ^q^	FS1	45.45 ± 0.54 ^jkl^	XY	42.42 ± 0.56 ^nop^	BB	40 ± 0.77 ^q^
QZP1	47.69 ± 0.66 ^ghij^	FS2	55.23 ± 0.56 ^b^	GX	43.85 ± 0.18 ^lmn^	LC	48.69 ± 0.62 ^gh^

Note: Values are expressed as means of three replicates ± SD. Different tiny letters in the same column indicate significant differences of *p* ≤ 0.01.

**Table 3 foods-11-02221-t003:** Fatty acid compositions of the 40 samples (%).

Sample Name	Oleic Acid (C18:1)	Linoleic Acid (C18:2)	Palmitic Acid (C16:0)	Stearic Acid (C18:0)	Linolenic Acid (C18:3)	Myristic Acid (C14:0)	Arachic Acid (C20:0)	Palmitoleic Acid (C16:1)	Unsaturated Fatty Acid (UFA)
QZ0	80.84 ± 1.88 ^bcdefg^	5.66 ± 0.37 ^ijklm^	9.09 ± 0.21 ^defgh^	3.44 ± 0.23 ^g^	0.18 ± 0.01 ^hij^	0.04 ± 0.01 ^bc^	0.43 ± 0.03 ^ij^	0.05 ± 0.01 ^efg^	86.73 ± 2.27 ^ab^
QZ1	70.71 ± 1.66 ^m^	3.95 ± 0.62 ^p^	7.94 ± 0.18 ^fgh^	2.59 ± 0.25 ^ijklmn^	0.19 ± 0.03 ^hij^	0.04 ± 0.00 ^bc^	0.48 ± 0.03 ^fghij^	0.04 ± 0.01 ^fg^	74.89 ± 2.32 ^fg^
QZ2	86.23 ± 0.59 ^a^	2.83 ± 0.12 ^q^	7.51 ± 0.12 ^gh^	2.58 ± 0.44 ^ijklmn^	0.15 ± 0.01 ^ij^	N.D.	0.48 ± 0.01 ^fghij^	0.03 ± 0.00 ^gh^	89.24 ± 0.72 ^ab^
QZ3	83.03 ± 1.48 ^abcd^	4.69 ± 0.21 ^no^	8.52 ± 0.06 ^fgh^	2.71 ± 0.27 ^ijklm^	0.18 ± 0.01 ^hij^	0.03 ± 0.01 ^bc^	0.51 ± 0.03 ^efghij^	0.05 ± 0.01 ^efg^	87.95 ± 1.71 ^ab^
QZ4	77.60 ± 0.62 ^ghij^	8.95 ± 0.51 ^c^	9.41 ± 0.21 ^defg^	2.83 ± 0.15 ^hijk^	0.26 ± 0.04 ^efghi^	0.03 ± 0.00 ^bc^	0.52 ± 0.04 ^efghij^	0.05 ± 0.01 ^efg^	86.86 ± 1.18 ^ab^
QZ5	80.53 ± 1.50 ^bcdefgh^	8.07 ± 0.35 ^d^	8.34 ± 0.18 ^fgh^	2.11 ± 0.12 ^no^	0.17 ± 0.02 ^hij^	0.02 ± 0.01 ^bc^	0.47 ± 0.03 ^fghij^	0.04 ± 0.01 ^fg^	88.81 ± 1.88 ^ab^
QZJ	82.24 ± 1.63 ^bcdef^	7.21 ± 0.16 ^ef^	7.49 ± 0.18 ^gh^	1.89 ± 0.09 ^o^	0.30 ± 0.04 ^defgh^	0.03 ± 0.01 ^bc^	0.47 ± 0.02 ^fghij^	0.09 ± 0.02 ^abc^	89.84 ± 1.85 ^a^
QZ8	79.78 ± 1.45 ^cdefghi^	4.62 ± 0.25 ^no^	11.74 ± 0.19 ^bcd^	2.87 ± 0.25 ^hij^	0.23 ± 0.03 ^efghij^	0.05 ± 0.02 ^bc^	0.44 ± 0.01 ^hij^	0.06 ± 0.01 ^def^	84.69 ± 1.74 ^bc^
QZ9	76.60 ± 1.57 ^hijk^	10.00 ± 0.19 ^b^	9.57 ± 0.36 ^defg^	2.59 ± 0.22 ^ijklmn^	0.21 ± 0.05 ^fghij^	0.05 ± 0.01 ^bc^	0.55 ± 0.03 ^efgh^	0.07 ± 0.02 ^cde^	86.88 ± 1.83 ^ab^
QZP1	82.80 ± 0.88 ^abcde^	5.50 ± 0.07 ^jklm^	7.94 ± 0.17 ^fgh^	2.72 ± 0.08 ^ijklm^	0.15 ± 0.01 ^ij^	0.03 ± 0.01 ^bc^	0.58 ± 0.05 ^def^	0.04 ± 0.00 ^fg^	88.49 ± 0.96 ^ab^
QZP2	82.20 ± 2.09 ^bcdef^	6.38 ± 0.27 ^gh^	8.28 ± 6.22 ^fgh^	2.29 ± 0.13 ^klmno^	0.18 ± 0.03 ^hij^	0.03 ± 0.00 ^bc^	0.48 ± 0.04 ^fghij^	0.04 ± 0.01 ^fg^	88.8 ± 2.40 ^ab^
NC1	81.29 ± 1.26 ^bcdefg^	7.13 ± 0.14 ^f^	8.35 ± 0.14 ^fghj^	1.77 ± 0.09 ^o^	0.32 ± 0.06 ^defg^	0.05 ± 0.01 ^bc^	0.54 ± 0.01 ^efghi^	0.11 ± 0.02 ^a^	88.85 ± 1.48 ^ab^
NC2	78.14 ± 0.32 ^fghij^	7.92 ± 0.4 ^d^	10.37 ± 0.15 ^cdef^	2.62 ± 0.19 ^ijklmn^	0.17 ± 0.02 ^hij^	0.04 ± 0.01 ^bc^	0.49 ± 0.04 ^efghij^	0.04 ± 0.00 ^fg^	86.27 ± 0.74 ^abc^
ND1	82.64 ± 1.37 ^abcde^	3.89 ± 0.14 ^p^	9.09 ± 0.34 ^defgh^	3.30 ± 0.12 ^gh^	0.20 ± 0.03 ^ghij^	0.03 ± 0.01 ^bc^	0.51 ± 0.02 ^efghij^	0.06 ± 0.01 ^def^	86.79 ± 1.55 ^ab^
ND3	82.77 ± 1.71 ^abcde^	5.25 ± 0.09 ^lmn^	8.25 ± 0.19 ^fgh^	2.60 ± 0.29 ^ijklmn^	0.22 ± 0.05 ^efghij^	0.04 ± 0.01 ^bc^	0.52 ± 0.03 ^efghij^	0.08 ± 0.02 ^bcd^	88.32 ± 1.87 ^ab^
HS1	79.58 ± 2.52 ^defghi^	7.76 ± 0.27 ^de^	9.55 ± 0.23 ^defg^	2.24 ± 0.21 ^lmno^	0.18 ± 0.05 ^hij^	0.03 ± 0.00 ^bc^	0.47 ± 0.04 ^fghij^	0.04 ± 0.00 ^fg^	87.56 ± 2.84 ^ab^
HS2	80.79 ± 1.68 ^bcdefg^	6.75 ± 0.29 ^fg^	8.74 ± 0.32 ^efgh^	2.90 ± 0.24 ^hij^	0.16 ± 0.02 ^ij^	0.03 ± 0.01 ^bc^	0.44 ± 0.08 ^hij^	0.04 ± 0.01 ^fg^	87.74 ± 2.00 ^ab^
HS3	83.73 ± 1.64 ^abc^	5.12 ± 0.08 ^mn^	7.61 ± 0.17 ^gh^	2.53 ± 0.12 ^jklmn^	0.17 ± 0.02 ^hij^	0.03 ± 0.01 ^bc^	0.50 ± 0.07 ^efghij^	0.04 ± 0.00 ^fg^	89.06 ± 1.74 ^ab^
FS1	81.09 ± 1.45 ^bcdefg^	6.10 ± 0.14 ^hijk^	9.13 ± 0.13 ^defgh^	2.70 ± 0.24 ^ijklm^	0.22 ± 0.04 ^efghij^	N.D.	0.47 ± 0.07 ^fghij^	N.D.	87.41 ± 1.63 ^ab^
FS2	82.81 ± 2.26 ^abcde^	4.10 ± 0.09 ^op^	9.00 ± 0.27 ^efgh^	3.12 ± 0.11 ^ghi^	0.22 ± 0.03 ^efghij^	0.04 ± 0.01 ^bc^	0.46 ± 0.07 ^ghij^	0.05 ± 0.01 ^efg^	87.18 ± 2.39 ^ab^
CM2	81.40 ± 1.81 ^bcdefg^	5.48 ± 0.35 ^klm^	9.30 ± 0.12 ^defg^	2.76 ± 0.12 ^ijkl^	0.18 ± 0.02 ^hij^	0.05 ± 0.01 ^bc^	0.56 ± 0.05 ^efg^	0.06 ± 0.01 ^def^	87.12 ± 2.19 ^ab^
CMDF	84.52 ± 2.61 ^ab^	3.95 ± 0.23 ^p^	7.94 ± 0.17 ^fgh^	2.59 ± 0.22 ^ijklmn^	0.19 ± 0.04 ^hij^	0.04 ± 0.01 ^bc^	0.48 ± 0.02 ^fghij^	0.04 ± 0.00 ^fg^	88.70 ± 2.88 ^ab^
QH	79.21 ± 0.78 ^defghi^	8.70 ± 0.26 ^c^	8.98 ± 0.37 ^efgh^	1.82 ± 0.14 ^o^	0.34 ± 0.04 ^de^	0.05 ± 0.02 ^bc^	0.55 ± 0.03 ^efgh^	0.09 ± 0.02 ^abc^	88.34 ± 1.10 ^ab^
HK	80.08 ± 1.16 ^cdefghi^	6.75 ± 0.32 ^fg^	10.05 ± 0.19 ^cdefg^	2.20 ± 0.13 ^mno^	0.16 ± 0.03 ^ij^	0.04 ± 0.01 ^bc^	0.51 ± 0.05 ^efghij^	0.05 ± 0.01 ^efg^	87.04 ± 1.52 ^ab^
BWL	79.89 ± 2.73 ^cdefghi^	6.14 ± 0.13 ^ghij^	9.63 ± 0.08 ^defg^	3.35 ± 0.14 ^gh^	0.22 ± 0.05 ^efghij^	0.03 ± 0.01 ^bc^	0.43 ± 0.04 ^ij^	0.06 ± 0.01 ^def^	86.31 ± 2.92 ^ab^
K13	66.34 ± 1.13 ^n^	5.28 ± 0.22 ^lmn^	13.54 ± 0.35 ^b^	8.87 ± 0.44 ^a^	0.87 ± 0.19 ^b^	0.62 ± 0.06 ^a^	0.60 ± 0.04 ^de^	N.D.	72.49 ± 1.54 ^g^
CR3	73.27 ± 1.33 ^klm^	5.04 ± 0.22 ^mn^	11.35 ± 0.36 ^bcde^	6.90 ± 0.26 ^b^	1.15 ± 0.12 ^a^	N.D.	0.67 ± 0.09 ^cd^	N.D.	79.46 ± 1.67 ^de^
CR	71.67 ± 1.62 ^lm^	7.03 ± 0.26 ^f^	13.06 ± 0.23 ^b^	3.99 ± 0.25 ^f^	0.40 ± 0.06 ^d^	N.D.	0.91 ± 0.09 ^b^	N.D.	79.10 ± 1.94 ^de^
XY	76.54 ± 2.42 ^ijk^	5.11 ± 0.25 ^mn^	9.68 ± 0.93 ^defg^	6.31 ± 0.14 ^c^	0.27 ± 0.03 ^efghi^	0.61 ± 0.07 ^a^	1.17 ± 0.06 ^a^	N.D.	81.92 ± 2.65 ^cd^
GX	81.53 ± 0.47 ^bcdefg^	5.25 ± 0.19 ^lmn^	9.16 ± 0.14 ^defgh^	3.02 ± 0.11 ^ghij^	0.21 ± 0.03 ^fghij^	0.03 ± 0.01 ^bc^	0.45 ± 0.03 ^ghij^	0.05 ± 0.01 ^efg^	87.04 ± 0.70 ^ab^
JX	81.63 ± 1.09 ^bcdefg^	6.15 ± 0.14 ^ghi^	9.23 ± 0.12 ^defg^	1.93 ± 0.25 ^o^	0.28 ± 0.05 ^efghi^	0.03 ± 0.00 ^bc^	0.44 ± 0.01 ^hij^	0.10 ± 0.02 ^ab^	88.16 ± 1.30 ^ab^
GZTR	80.91 ± 1.56 ^bcdefg^	5.79 ± 0.24 ^hijkl^	9.14 ± 0.22 ^defgh^	3.12 ± 0.18 ^ghi^	0.20 ± 0.02 ^ghij^	0.03 ± 0.01 ^bc^	0.47 ± 0.05 ^fghij^	0.06 ± 0.01 ^def^	86.96 ± 1.83 ^ab^
HN	78.69 ± 2.23 ^efghi^	7.82 ± 0.44 ^de^	9.56 ± 0.23 ^defg^	2.90 ± 0.16 ^hij^	0.22 ± 0.05 ^efghij^	0.04 ± 0.01 ^bc^	0.45 ± 0.04 ^ghij^	0.05 ± 0.01 ^efg^	86.78 ± 2.73 ^ab^
DA	74.72 ± 2.23 ^jkl^	10.94 ± 0.21 ^a^	11.19 ± 0.18 ^bcde^	1.92 ± 0.27 ^o^	0.33 ± 0.05 ^def^	0.06 ± 0.02 ^b^	0.50 ± 0.06 ^efghij^	0.08 ± 0.01 ^bcd^	86.07 ± 2.50 ^abc^
WC	80.56 ± 1.12 ^bcdefgh^	5.16 ± 0.15 ^lmn^	9.72 ± 0.37 ^defg^	3.53 ± 0.12 ^fg^	0.23 ± 0.04 ^efghij^	0.04 ± 0.01 ^bc^	0.41 ± 0.01 ^j^	0.06 ± 0.01 ^def^	86.01 ± 1.32 ^abc^
WH	83.1 ± 1.47 ^abcd^	4.01 ± 0.32 ^p^	6.55 ± 0.31 ^h^	5.15 ± 0.19 ^e^	0.12 ± 0.02 ^j^	0.02 ± 0.01 ^bc^	0.52 ± 0.03 ^efghij^	0.03 ± 0.00 ^gh^	87.26 ± 1.81 ^ab^
BT	1.23 ± 0.07 ^p^	0.69 ± 0.03 ^r^	0.44 ± 0.04 ^i^	0.11 ± 0.01 ^p^	0.01 ± 0.00 ^k^	0.01 ± 0.00 ^c^	0.02 ± 0.00 ^k^	0.01 ± 0.00 ^h^	1.94 ± 0.10 ^h^
GZ	78.69 ± 1.11 ^efghi^	7.82 ± 0.29 ^de^	9.56 ± 0.15 ^defg^	2.90 ± 0.17 ^hij^	0.22 ± 0.04 ^efghij^	0.04 ± 0.01 ^bc^	0.45 ± 0.02 ^ghij^	0.05 ± 0.01 ^efg^	86.78 ± 1.45 ^ab^
BB	62.22 ± 0.99 ^o^	10.85 ± 0.28 ^a^	17.76 ± 0.54 ^a^	5.65 ± 0.32 ^d^	0.86 ± 0.05 ^b^	N.D.	0.75 ± 0.04 ^c^	N.D.	73.93 ± 1.32 ^fg^
LC	71.21 ± 0.78 ^lm^	5.63 ± 0.29 ^ijklm^	12.24 ± 0.23 ^bc^	6.68 ± 0.33 ^bc^	0.53 ± 0.03 ^c^	N.D.	0.74 ± 0.06 ^c^	N.D.	77.37 ± 1.10 ^ef^

Note: Values are expressed as means of three replicates ± SD. Different tiny letters in the same column indicate significant differences of *p* ≤ 0.01. N.D.: not detected. UFA (C18:1 + C18:2 + C18:3 + C16:1).

**Table 4 foods-11-02221-t004:** Correlation of fatty acid composition (Pearson’s correlation coefficient).

	Oleic Acid	Linoleic Acid	Palmitic Acid	Stearic Acid
oleic acid	1	−0.919 **	−0.825 **	0.294
linoleic acid		1	0.606 **	−0.516 **
palmitic acid			1	−0.283
stearic acid				1

Note: ** significant at level 0.01.

**Table 5 foods-11-02221-t005:** Comprehensive score and ranking of 40 kinds of *Camellia* spp. quality.

Sample Name	Major Constituent 1 Score	Major Constituent 2 Score	Major Constituent 3 Score	Major Constituent 4 Score	Major Constituent 5 Score	Comprehensive Score	Sort
QZ0	0.11	−0.17	0.97	0.33	−0.04	0.15	17
QZ1	1.17	1.71	−0.30	0.49	0.02	0.95	3
QZ2	0.41	−0.73	0.70	0.34	2.80	0.36	9
QZ3	0.43	−0.60	−0.02	0.44	−1.07	−0.04	19
QZ4	0.42	0.17	−0.77	0.45	1.10	0.27	13
QZ5	−0.11	−0.37	0.77	−0.42	−0.85	−0.16	22
QZJ	0.45	−0.43	0.23	0.82	−0.24	0.17	16
QZ8	0.50	0.23	0.04	1.06	−0.06	0.37	8
QZ9	0.69	0.18	−0.62	0.75	0.02	0.33	11
QZP1	0.21	−1.01	0.18	0.09	−0.27	−0.17	23
QZP2	−0.24	−0.49	0.20	−0.12	0.29	−0.19	25
NC1	0.24	−1.01	0.35	−0.01	−0.40	−0.16	21
NC2	0.49	0.49	−1.30	0.77	−0.53	0.19	15
ND1	−0.18	−1.15	−0.32	−0.60	−0.82	−0.56	35
ND3	0.03	−0.95	−0.20	0.01	−1.54	−0.40	29
HS1	0.47	0.44	−0.20	0.54	0.08	0.35	10
HS2	0.42	−1.52	2.25	0.08	2.78	0.33	12
HS3	1.14	1.04	0.10	0.85	−0.59	0.80	4
FS1	−0.06	−1.35	0.35	−0.14	−0.72	−0.41	31
FS2	0.37	1.48	0.26	−1.39	0.76	0.51	6
CM2	−0.36	−0.10	2.75	−1.57	−2.16	−0.18	24
CMDF	1.54	1.60	1.04	1.65	−0.49	1.32	1
QH	0.91	0.73	−0.08	1.02	−0.92	0.58	5
HK	−0.21	−0.60	−0.67	−0.22	−0.56	−0.41	30
BWL	0.41	−0.99	1.07	0.51	−0.74	0.03	18
K13	−2.09	0.41	−0.06	0.61	1.39	−0.59	36
CR3	−1.59	0.62	1.43	−0.69	−0.55	−0.44	32
CR	−2.02	1.19	−0.01	0.61	1.76	−0.32	28
XY	−2.27	0.50	−1.09	0.58	0.53	−0.86	40
GX	−0.09	−0.33	−1.29	−0.06	0.23	−0.28	27
JX	−0.01	−0.50	−0.69	0.07	0.06	−0.21	26
GZTR	−0.40	−0.93	−1.64	−0.58	0.78	−0.61	38
HN	0.37	0.62	−1.00	0.62	0.01	0.25	14
DA	0.58	0.26	0.93	0.11	−0.40	0.41	7
WC	0.13	−0.82	0.87	0.09	−0.08	−0.05	20
WH	0.15	−0.55	−1.53	−4.32	0.78	−0.63	39
BT	−0.02	−0.60	−2.28	−2.05	0.60	−0.61	37
GZ	1.19	1.13	0.95	0.31	0.06	0.95	2
BB	−1.76	1.45	−1.50	0.35	−0.32	−0.54	34
LC	−1.43	0.96	0.13	−1.36	−0.70	−0.53	33

## Data Availability

The data presented in this study are available on request from the corresponding author. The data are not publicly available due to privacy.

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
