# Peer review of "Quality Evaluation of the Oil of Camellia spp."

_foods, 2022, doi:10.3390/foods11152221_

Round 1

Reviewer 1 Report

This study shows results from 40 samples of camellia seeds using different analysis methods in order to evaluate their potential for further breeding. The natural variability of camellia seed oil is of high importance as this oil covers a growing market share. However, some results of this study are only shown in small diagramms with very limited information. Therefore, a revision of some parts of the figures and/or a change into a table should be considered. The readability of some characters in figure 2 to 4 are not sufficient.

Detailed comments:

1-2        oil of Camellia (from my point of view)

12          The oil of Camellia oleifera spp. has become a well-kown … oil (not the tree is the subject).

15-16    …40 kinds of C. oleifera spp. and C. vietnamensis seeds… (add the other varieties in order to cover all seeds analysed)

23-24    Which new method was provided by this study? (Perhaps better: these methods were applied for the first time? However, I do not know, whether this is correct.)

26          “bioactive content” perhaps better “minor compounds”? (Which compound is not bioactive?); fatty acid composition (to be added to the key words); “quality” Which kind of quality do you mean? Please specify in more detail; Perhaps, the other applied analytical methods and statistics should be mentioned here.

40-42   Delete the claimed beneficial effects for oleic acid …”, which can bate blood-vessel and prevent cardiovascular and cerebrovascular diseases,“... Otherwise, these claims have to be proven by references.

42          …It is known as a “safe fatty acid”… An article in a scientific journal should be concise and precise. Do not use “…” to indicate unclear messages. Delete this sentence or clarify your intention. What is your exact message? Oleic acid has a status as generally recognized as safe (GRAS)?

49-50    …highest quality oils… A quality has to be defined in order to state a significant message. Please, clarify the specific quality parameters, which are best/highest for this oil. Otherwise, delete …”, and is considered to be one of the highest quality oils“.

54-56    …isolated….from…seeds. I would assume that the percentages of fatty acids are with regard to the oil not to the seeds. Please revise the sentence.

72 Who provided the seed samples? Is there any botanical justification about the different claimed varieties (for instance Quongzhong 0 to 9)? Or were these samples different accessions of the same variety? This would be of significance for the reader. In addition, specify the year of harvest. Please, clarify.

85 I would assume that the seeds were not shelled and crushed in an oven? Please, clarify or revise the sentence.

87 to obtain a deoiled powder

102-104 The gas flow rates point to the use of a FID. The use of a transfer line and an ion source temperature points to the use of an MSD. Please clarify, what has been used for FAME analysis.

108 Please specify what is "positive hexane".

114 Please specify what do you mean: “without diversion”. Do you mean splitless?

116-119 As no standards were used for identification please add “tentatively identified”

120 see above (87)

190 figure1 gives only a rough estimation about the oil contents. A table would provide precise data. Foods is a scientific journal and readers interested in this topic would appreciate a table. Please, consider to change figure 1 into a table.

199 The authors used common fatty acid names except for octadecanoic acid. So this should be mentioned as stearic acid? Please clarify.

203 arachidic acid (C20:0) is not an unsaturated fatty acid, but has to be counted into the saturated fatty acids.

205 To characterise erucic acid as docosa-13-enoic acid would be more specific. Please, revise.

206 What is the meaning of “colleseed oil digestion”? Please clarify. Do you mean high erucic rapeseed oil or mustard seed oil?

209 Figure 2 gives no indication about the erucic acid content of camellia oils. However, it is summarized that excluding “mustard acid” (why a new name here?) camellia oils is nutritious, but also in the text no content of erucic acid has been reported. Please add a complete table about your fatty acid composition results. This would increase a lot the impact of your contribution.

213 Figure2: For FAME composition it is the same as for oil content. From my point of view a table would be much better. Fatty acid names in the figure are not the same as in the text, please revise. What is UFA? The legend is not complete. For gaidic acid (as specified in the figure) a IUPAC Name of (E)-hexadec-2-enoic acid can be found. Is this fatty acid really present in Camellia oil? This would be very uncommon. Please clarify.

Further on, figure 2 A and B were discussed later on in the article. Transfer this part of the figure to the matching paragraph.

225 Table 2: specify the meaning of the asterisks.  What is your conclusion about this analysis? Are there any consequences, benefits or drawbacks? What is the intention for this mathematical analysis?

227 The term “bioactive components” is not precise. Fatty acids are also bioactive components, but reported in the section above. From my point of view the following components are minor compounds from the oil or polar compounds from the extraction meal.

229 The relative peak area of the total ion chromatogram (TIC) of tentatively identified components of different samples were compared. No content was determined from my point of view. Please revise the sentence.

230 Vitamin E is a name for a group of substances. Please specify in detail, which tocopherol was meant.  

230 You only mention stigmasterol, but not β-sitosterol. As β-sitosterol is the most important sterol in most edible oils, it would be of importance to mention any difference regarding camellia seed oil. Please discuss in more detail.

232-234 In the case that these effects are the same as in the introduction, this can be abbreviated. For new claims there should be a reference.

240 Figure 3: Please specify in more detail, how you evaluated the GC-MS chromatogram. According to the procedure the oil was diluted, filtered and injected. However, the oven program of the GC run was stopped after reaching 270 °C. What happened with the triacylglycerols injected into the analysis system? In the diagram C of the TIC we see relative peak areas of up to 10 % for stigmasterol, 15 % for tocopherols (without silylation?), squalene up to more than 20 % and β-amyrin also up to more than 20 %. As these minor compounds are normaly detected at a very small level, there should be a lot of peaks disregarded. Please specify in more detail the used procedure.

Reference 35 is not significant for the statement in the sentence as it deals with the analysis of triacylglycerols using HPLC. Please refer to an adequate refence. In addition, the internet address lead to a completely differing article.

Author Response

Dear reviewer,

Thank you very much for forwarding the reviewers’ comments and giving us an opportunity to improve our manuscript (ID: 1818153). All suggestions have been carefully considered and the comments have been addressed point by point. The manuscript has been revised accordingly in red font. The responses to the reviewers’ comments are given below.

Responses addressing Reviewers’ Comments

Reviewer #1: 

Comment 1: 1-2 oil of Camellia (from my point of view).

Response:

Thank you for the valuable suggestion. In accordance with the Reviewer’s comment, we have made correction on line 2 in the revised manuscript and are marked in red.

Comment 2: 12 The oil of Camellia oleifera spp. has become a well-kown … oil (not the tree is the subject). 

Response:

Thank you for the valuable suggestion. In accordance with the Reviewer’s comment, we have made corrections on line 12 in the revised manuscript, marked in red.

Comment 3: 15-16 …40 kinds of C. oleifera spp. and C. vietnamensis seeds… (add the other varieties in order to cover all seeds analysed).

Response:

Thank you for the valuable suggestion. Considering the reviewer’s suggestion, we have made correction on line 16-17 in the revised manuscript and are marked in red.

Comment 4: 23-24 Which new method was provided by this study? (Perhaps better: these methods were applied for the first time? However, I do not know, whether this is correct.)

Response:

Thank you for the valuable suggestion and we apologize for the unclear language expression. These methods were applied for the first time for the quality evaluation of Camellia spp. species. In accordance with the Reviewer’s comments, we have have made correction on line 24-26 in the revised manuscript, marked in red.

Comment 5: 26 “bioactive content” perhaps better “minor compounds”? (Which compound is not bioactive?); fatty acid composition (to be added to the key words); “quality” Which kind of quality do you mean? Please specify in more detail; Perhaps, the other applied analytical methods and statistics should be mentioned here.

Response:

Thank you for this valuable suggestion. We have changed“bioactive content”to “minor compounds” and added “fatty acid composition” as the key words. The “quality” contained the fatty acid composition, minor compounds, and antioxidant activity of C. oleifera spp. Through these, the quality of different kinds of C. oleifera spp. was evaluated. In accordance with the Reviewer’s comments, we have have made correction on line 28 in the revised manuscript, marked in red.

Comment 6: 40-42 Delete the claimed beneficial effects for oleic acid …”, which can bate blood-vessel and prevent cardiovascular and cerebrovascular diseases,“... Otherwise, these claims have to be proven by references.

Response:

Thank you for this valuable suggestion and we are very sorry for our negligence regarding the citations. The reference of these claims about oleic acid was the seventh reference ([7]). According to the reviewer’s comment, we have made correction on line 45 in the revised manuscript and are marked in red.

Reference

[7] Zhong, H.Y.; Bedgood, D.R.; Bishop A.G.,; Prenzler P.D.; Robards K. Effect of added caffeic acid and tyrosol on the fatty acid and volatile profiles of camellia oil following heating. Journal of Agricultural and Food Chemistry. 2006, 54 (25), 9551-9558. https://doi.org/10.1021/jf061974z.

Comment 7: 42 …It is known as a “safe fatty acid”… An article in a scientific journal should be concise and precise. Do not use “…” to indicate unclear messages. Delete this sentence or clarify your intention. What is your exact message? Oleic acid has a status as generally recognized as safe (GRAS)?

Response:

Thank you for this valuable suggestion and we apologize for the unclear language expression. What we want to express is that oleic acid has a status as generally recognized as safe. According to the reviewer’s comment, we have deleted this sentence on line 45 in the revised manuscript and are marked in red.

Comment 8: 49-50 …highest quality oils… A quality has to be defined in order to state a significant message. Please, clarify the specific quality parameters, which are best/highest for this oil. Otherwise, delete …”, and is considered to be one of the highest quality oils“.

Response:

Thank you for this valuable suggestion and we apologize for the unclear language expression. According to the reviewer’s comment, we have made correction on line 57 in the revised manuscript and are marked in red.

Comment 9: 54-56 …isolated….from…seeds. I would assume that the percentages of fatty acids are with regard to the oil not to the seeds. Please revise the sentence.

Response:

Thank you for this valuable suggestion and we are very sorry for our negligence regarding the citations. According to the reviewer’s comment, we have made correction on line 61-63 in the revised manuscript and are marked in red.

Comment 10: 72 Who provided the seed samples? Is there any botanical justification about the different claimed varieties (for instance Quongzhong 0 to 9)? Or were these samples different accessions of the same variety? This would be of significance for the reader. In addition, specify the year of harvest. Please, clarify.

Response:

Thank you for this valuable suggestion. The seed samples were harvested in 2018 and provided by Prof. Kaibing Zhou. Qiongzhong 0 to 9 were different accessions of the same species. According to the reviewer’s comment, we have made correction on line 80-81 in the revised manuscript and are marked in red.

Comment 11: 85 I would assume that the seeds were not shelled and crushed in an oven? Please, clarify or revise the sentence.

Response:

Thank you for this valuable suggestion and we apologize for the unclear language expression. The seeds were dried in the oven at 50 ℃, then shelled and crushed. According to the reviewer’s comment, we have made correction on line 94 in the revised manuscript and are marked in red.

Comment 12: 87 to obtain a deoiled powder

Response:

Thank you for this valuable suggestion and we apologize for the unclear language expression. According to the reviewer’s comment, we have made correction on line 97 in the revised manuscript and are marked in red.

Comment 13: 102-104 The gas flow rates point to the use of a FID. The use of a transfer line and an ion source temperature points to the use of an MSD. Please clarify, what has been used for FAME analysis.

Response:

Thank you for this valuable suggestion and we are very sorry for our negligence regarding the citations. We detected the fatty acid composition by GC and the gas flow rates used the FID. There was the detector temperature setting 270 ℃ instead of the temperature of the transfer line and ion source. According to the reviewer’s comment, we have made correction on lines 111-112 in the revised manuscript and are marked in red.

Comment 14: 108 Please specify what is "positive hexane".

Response:

Thank you for this valuable suggestion and we apologize for the unclear language expression. The "positive hexane" is n-hexane. According to the reviewer’s comment, we have made correction on line 118 in the revised manuscript and are marked in red.

Comment 15: 114 Please specify what do you mean: “without diversion”. Do you mean splitless?

Response:

Thank you for this valuable suggestion and we apologize for the unclear language expression. Yes, it means splitless. According to the reviewer’s comment, we have made correction on line 124 in the revised manuscript and are marked in red.

Comment 16: 116-119 As no standards were used for identification please add “tentatively identified”.

Response:

Thank you for this valuable suggestion and we apologize for the unclear language expression. According to the reviewer’s comment, we have made correction on line 130 in the revised manuscript and are marked in red.

Comment 17: 120 see above (87).

Response:

Thanks for your kind suggestion and we apologize for the unclear language expression. According to the reviewer’s comment, we have changed “degreasing powder” to “deoiled powder” on line 133 in the revised manuscript and are marked in red.

Comment 18: 190 figure1 gives only a rough estimation about the oil contents. A table would provide precise data. Foods is a scientific journal and readers interested in this topic would appreciate a table. Please, consider to change figure 1 into a table.

Response:

Thank you for this valuable suggestion. According to the reviewer’s comment, we have changed figure 1 into a table (Table 2) on line 210-212 in the revised manuscript and are marked in red.

Table 2  The oil content of the different samples.

Sample name

oil(w/w: %)

Sample name

oil(w/w: %)

Sample name

oil(w/w: %)

Sample name

oil(w/w: %)

QZ0

48.19±0.96 ghi

QZP2

45.92±0.31 ijkl

CM2

47.64±0.69 ghij

JX

44.14±0.52 lmn

QZ1

52.81±0.82 cd

NC1

54.46±1.42 bc

CMDF

58.96±0.58 a

GZTR

44.01±0.17 lmn

QZ2

45.21±0.65 klm

NC2

46.82±2.15 hijk

QH

45.46±1.09 jkl

HN

40.81±1.37 opq

QZ3

45.12±0.89 klm

ND1

49.78±2.54 fg

HK

43.98±1.31 lmn

DA

46.08±0.71 ijkl

QZ4

51.72±1.66 def

ND3

58.77±1.08 a

BWL

51.52±0.56 def

WC

48.11±0.43 ghi

QZ5

41.35±0.37 opq

HS1

53.74±0.68 bcd

K13

40.18±0.61 pq

WH

41.2±0.33 opq

QZJ

47.82±1.29 ghi

HS2

53±0.72 cd

CR3

44.56±1.29 klmn

BT

30.22±1.16 r

QZ8

52.1±0.24 de

HS3

49.88±0.82 efg

CR

42.91±0.58 mno

GZ

45.04±0.67 klm

QZ9

40.01±0.53 q

FS1

45.45±0.54 jkl

XY

42.42±0.56 nop

BB

40±0.77 q

QZP1

47.69±0.66 ghij

FS2

55.23±0.56 b

GX

43.85±0.18 lmn

LC

48.69±0.62 gh

Note: Values are expressed as means of three replicates ± SD. Different tiny letters in the same column indicate significant differences of p ≤ 0.01.

Comment 19: 199 The authors used common fatty acid names except for octadecanoic acid. So this should be mentioned as stearic acid? Please clarify.

Response:

Thanks for your kind suggestion and we apologize for the unclear language expression. According to the reviewer’s comment, we have changed octadecanoic acid to stearic acid on line 221 and in Table 3 in the revised manuscript and are marked in red.

Comment 20: 203 arachidic acid (C20:0) is not an unsaturated fatty acid, but has to be counted into the saturated fatty acids.

Response:

Thanks for your kind suggestion and we are very sorry for our negligence regarding the citations. According to the reviewer’s comment, we have made correction on line 226 in the revised manuscript and are marked in red.

Comment 21: 205 To characterise erucic acid as docosa-13-enoic acid would be more specific. Please, revise.

Response:

Thank you for this valuable suggestion. According to the reviewer’s comment, we have made correction on line 379, 380, 383, 384 in the revised manuscript and are marked in red.

Comment 22: 206 What is the meaning of “colleseed oil digestion”? Please clarify. Do you mean high erucic rapeseed oil or mustard seed oil?

Response:

Thank you for this valuable suggestion. We mean that docosa-13-enoic acid may affect the digestion of rapeseed oil in humans, then cause myocardial damage, make the cholesterol level of adrenal tissue rise, and easily make fat accumulate in the heart tissue. According to the reviewer’s comment, we have made correction on line 380-381 in the revised manuscript and are marked in red.

Comment 23: 209 Figure 2 gives no indication about the erucic acid content of camellia oils. However, it is summarized that excluding “mustard acid” (why a new name here?) camellia oils is nutritious, but also in the text no content of erucic acid has been reported. Please add a complete table about your fatty acid composition results. This would increase a lot the impact of your contribution.

Response:

Thanks for your kind suggestion and we apologize for the unclear language expression. We wanted to discuss the difference in docosa-13-enoic acid between camellia oil and rapeseed oil, where no docosa-13-enoic acid was detected in camellia oil in this study, while rapeseed oil contains docosa-13-enoic acid that damages the heart muscle. We put lines 205-212 into the discussion section on line 379-387. According to the reviewer’s comment, we have made correction on line `379-387 in the revised manuscript and are marked in red.

Comment 24: 213 Figure2: For FAME composition it is the same as for oil content. From my point of view a table would be much better. Fatty acid names in the figure are not the same as in the text, please revise. What is UFA? The legend is not complete. For gaidic acid (as specified in the figure) a IUPAC Name of (E)-hexadec-2-enoic acid can be found. Is this fatty acid really present in Camellia oil? This would be very uncommon. Please clarify. 

Further on, figure 2 A and B were discussed later on in the article. Transfer this part of the figure to the matching paragraph.

Response:

Thank you for this valuable suggestion. We have changed FAME composition of figure 2 into a table (Table 3). Fatty acid names in the figure have changed to the same as in the text. The UFA is unsaturated fatty acids. We detected the palmitoleic acid (2-hexadecenoic acid) in Camellia oil, that was gaidic acid in the figure. For figure 2 A and B, we have transferred them to the matching paragraph (Figure 3A and B). According to the reviewer’s comment, we have made correction on line 228-231 and line 341-344 in the revised manuscript and are marked in red.

Table 3

Fatty acid compositions of the fourty kinds of Camellia oleifera. (%)

Sample name

oleic acid (C18:1)

linoleic acid (C18:2)

palmitic acid (C16:0)

stearic acid (C18:0)

linolenic acid (C18:3)

myristic acid (C14:0)

arachic acid (C20:0)

palmitoleic acid (C16:1)

unsaturated fatty acid (UFA)

QZ0

80.84±1.88 bcdefg

5.66±0.37 ijklm

9.09±0.21 defgh

3.44±0.23 g

0.18±0.01 hij

0.04±0.01 bc

0.43±0.03 ij

0.05±0.01 efg

86.73±2.27 ab

QZ1

70.71±1.66 m

3.95±0.62 p

7.94±0.18 fgh

2.59±0.25 ijklmn

0.19±0.03 hij

0.04±0.00 bc

0.48±0.03 fghij

0.04±0.01 fg

74.89±2.32 fg

QZ2

86.23±0.59 a

2.83±0.12 q

7.51±0.12 gh

2.58±0.44 ijklmn

0.15±0.01 ij

N.D.

0.48±0.01 fghij

0.03±0.00 gh

89.24±0.72 ab

QZ3

83.03±1.48 abcd

4.69±0.21 no

8.52±0.06 fgh

2.71±0.27 ijklm

0.18±0.01 hij

0.03±0.01 bc

0.51±0.03 efghij

0.05±0.01 efg

87.95±1.71 ab

QZ4

77.60±0.62 ghij

8.95±0.51 c

9.41±0.21 defg

2.83±0.15 hijk

0.26±0.04 efghi

0.03±0.00 bc

0.52±0.04 efghij

0.05±0.01 efg

86.86±1.18 ab

QZ5

80.53±1.50 bcdefgh

8.07±0.35 d

8.34±0.18 fgh

2.11±0.12 no

0.17±0.02 hij

0.02±0.01 bc

0.47±0.03 fghij

0.04±0.01 fg

88.81±1.88 ab

QZJ

82.24±1.63 bcdef

7.21±0.16 ef

7.49±0.18 gh

1.89±0.09 o

0.30±0.04 defgh

0.03±0.01 bc

0.47±0.02 fghij

0.09±0.02 abc

89.84±1.85 a

QZ8

79.78±1.45 cdefghi

4.62±0.25 no

11.74±0.19 bcd

2.87±0.25 hij

0.23±0.03 efghij

0.05±0.02 bc

0.44±0.01 hij

0.06±0.01 def

84.69±1.74 bc

QZ9

76.60±1.57 hijk

10.00±0.19 b

9.57±0.36 defg

2.59±0.22 ijklmn

0.21±0.05 fghij

0.05±0.01 bc

0.55±0.03 efgh

0.07±0.02 cde

86.88±1.83 ab

QZP1

82.80±0.88 abcde

5.50±0.07 jklm

7.94±0.17 fgh

2.72±0.08 ijklm

0.15±0.01 ij

0.03±0.01 bc

0.58±0.05 def

0.04±0.00 fg

88.49±0.96 ab

QZP2

82.20±2.09 bcdef

6.38±0.27 gh

8.28±6.22 fgh

2.29±0.13 klmno

0.18±0.03 hij

0.03±0.00 bc

0.48±0.04 fghij

0.04±0.01 fg

88.8±2.40 ab

NC1

81.29±1.26 bcdefg

7.13±0.14 f

8.35±0.14 fghj

1.77±0.09 o

0.32±0.06 defg

0.05±0.01 bc

0.54±0.01 efghi

0.11±0.02 a

88.85±1.48 ab

NC2

78.14±0.32 fghij

7.92±0.4 d

10.37±0.15 cdef

2.62±0.19 ijklmn

0.17±0.02 hij

0.04±0.01 bc

0.49±0.04 efghij

0.04±0.00 fg

86.27±0.74 abc

ND1

82.64±1.37 abcde

3.89±0.14 p

9.09±0.34 defgh

3.30±0.12 gh

0.20±0.03 ghij

0.03±0.01 bc

0.51±0.02 efghij

0.06±0.01 def

86.79±1.55 ab

ND3

82.77±1.71 abcde

5.25±0.09 lmn

8.25±0.19 fgh

2.60±0.29 ijklmn

0.22±0.05 efghij

0.04±0.01 bc

0.52±0.03 efghij

0.08±0.02 bcd

88.32±1.87 ab

HS1

79.58±2.52 defghi

7.76±0.27 de

9.55±0.23 defg

2.24±0.21 lmno

0.18±0.05 hij

0.03±0.00 bc

0.47±0.04 fghij

0.04±0.00 fg

87.56±2.84 ab

HS2

80.79±1.68 bcdefg

6.75±0.29 fg

8.74±0.32 efgh

2.90±0.24 hij

0.16±0.02 ij

0.03±0.01 bc

0.44±0.08 hij

0.04±0.01 fg

87.74±2.00 ab

HS3

83.73±1.64 abc

5.12±0.08 mn

7.61±0.17 gh

2.53±0.12 jklmn

0.17±0.02 hij

0.03±0.01 bc

0.50±0.07 efghij

0.04±0.00 fg

89.06±1.74 ab

FS1

81.09±1.45 bcdefg

6.10±0.14 hijk

9.13±0.13 defgh

2.70±0.24 ijklm

0.22±0.04 efghij

N.D.

0.47±0.07 fghij

N.D.

87.41±1.63 ab

FS2

82.81±2.26 abcde

4.10±0.09 op

9.00±0.27 efgh

3.12±0.11 ghi

0.22±0.03 efghij

0.04±0.01 bc

0.46±0.07 ghij

0.05±0.01 efg

87.18±2.39 ab

CM2

81.40±1.81 bcdefg

5.48±0.35 klm

9.30±0.12 defg

2.76±0.12 ijkl

0.18±0.02 hij

0.05±0.01 bc

0.56±0.05 efg

0.06±0.01 def

87.12±2.19 ab

CMDF

84.52±2.61 ab

3.95±0.23 p

7.94±0.17 fgh

2.59±0.22 ijklmn

0.19±0.04 hij

0.04±0.01 bc

0.48±0.02 fghij

0.04±0.00 fg

88.70±2.88 ab

QH

79.21±0.78 defghi

8.70±0.26 c

8.98±0.37 efgh

1.82±0.14 o

0.34±0.04 de

0.05±0.02 bc

0.55±0.03 efgh

0.09±0.02 abc

88.34±1.10 ab

HK

80.08±1.16 cdefghi

6.75±0.32 fg

10.05±0.19 cdefg

2.20±0.13 mno

0.16±0.03 ij

0.04±0.01 bc

0.51±0.05 efghij

0.05±0.01 efg

87.04±1.52 ab

BWL

79.89±2.73 cdefghi

6.14±0.13 ghij

9.63±0.08 defg

3.35±0.14 gh

0.22±0.05 efghij

0.03±0.01 bc

0.43±0.04 ij

0.06±0.01 def

86.31±2.92 ab

K13

66.34±1.13 n

5.28±0.22 lmn

13.54±0.35 b

8.87±0.44 a

0.87±0.19 b

0.62±0.06 a

0.60±0.04 de

N.D.

72.49±1.54 g

CR3

73.27±1.33 klm

5.04±0.22 mn

11.35±0.36 bcde

6.90±0.26 b

1.15±0.12 a

N.D.

0.67±0.09 cd

N.D.

79.46±1.67 de

CR

71.67±1.62 lm

7.03±0.26 f

13.06±0.23 b

3.99±0.25 f

0.40±0.06 d

N.D.

0.91±0.09 b

N.D.

79.10±1.94 de

XY

76.54±2.42 ijk

5.11±0.25 mn

9.68±0.93 defg

6.31±0.14 c

0.27±0.03 efghi

0.61±0.07 a

1.17±0.06 a

N.D.

81.92±2.65 cd

GX

81.53±0.47 bcdefg

5.25±0.19 lmn

9.16±0.14 defgh

3.02±0.11 ghij

0.21±0.03 fghij

0.03±0.01 bc

0.45±0.03 ghij

0.05±0.01 efg

87.04±0.70 ab

JX

81.63±1.09 bcdefg

6.15±0.14 ghi

9.23±0.12 defg

1.93±0.25 o

0.28±0.05 efghi

0.03±0.00 bc

0.44±0.01 hij

0.10±0.02 ab

88.16±1.30 ab

GZTR

80.91±1.56 bcdefg

5.79±0.24 hijkl

9.14±0.22 defgh

3.12±0.18 ghi

0.20±0.02 ghij

0.03±0.01 bc

0.47±0.05 fghij

0.06±0.01 def

86.96±1.83 ab

HN

78.69±2.23 efghi

7.82±0.44 de

9.56±0.23 defg

2.90±0.16 hij

0.22±0.05 efghij

0.04±0.01 bc

0.45±0.04 ghij

0.05±0.01 efg

86.78±2.73 ab

DA

74.72±2.23 jkl

10.94±0.21 a

11.19±0.18 bcde

1.92±0.27 o

0.33±0.05 def

0.06±0.02 b

0.50±0.06 efghij

0.08±0.01 bcd

86.07±2.50 abc

WC

80.56±1.12 bcdefgh

5.16±0.15 lmn

9.72±0.37 defg

3.53±0.12 fg

0.23±0.04 efghij

0.04±0.01 bc

0.41±0.01 j

0.06±0.01 def

86.01±1.32 abc

WH

83.1±1.47 abcd

4.01±0.32 p

6.55±0.31 h

5.15±0.19 e

0.12±0.02 j

0.02±0.01 bc

0.52±0.03 efghij

0.03±0.00 gh

87.26±1.81 ab

BT

1.23±0.07 p

0.69±0.03 r

0.44±0.04 i

0.11±0.01 p

0.01±0.00 k

0.01±0.00 c

0.02±0.00 k

0.01±0.00 h

1.94±0.10 h

GZ

78.69±1.11 efghi

7.82±0.29 de

9.56±0.15 defg

2.90±0.17 hij

0.22±0.04 efghij

0.04±0.01 bc

0.45±0.02 ghij

0.05±0.01 efg

86.78±1.45 ab

BB

62.22±0.99 o

10.85±0.28 a

17.76±0.54 a

5.65±0.32 d

0.86±0.05 b

N.D.

0.75±0.04 c

N.D.

73.93±1.32 fg

LC

71.21±0.78 lm

5.63±0.29 ijklm

12.24±0.23 bc

6.68±0.33 bc

0.53±0.03 c

N.D.

0.74±0.06 c

N.D.

77.37±1.10 ef

Note: Values are expressed as means of three replicates ± SD. Different tiny letters in the same column indicate significant differences of p ≤ 0.01. N.D.: not detected. UFA (C18:1 + C18:2 + C18:3 + C16:1).

Figure 3. (A): The heatmap of fatty acids in Camellia spp., respectively; (B): The PCA analysis of fatty acids in Camellia spp., respectively; (C): The heatmap of active content in Camellia spp., respectively; (D): The PCA analysis of active content in Camellia spp., respectively.

Comment 25: 225 Table 2: specify the meaning of the asterisks. What is your conclusion about this analysis? Are there any consequences, benefits or drawbacks? What is the intention for this mathematical analysis?

Response:

Thank you for this valuable suggestion and we are very sorry for our negligence regarding the citations. We have supplemented the note: * * significant at level 0.01; * significant at level 0.05. The results show that the oleic acid content had a significant negative correlation with the linoleic acid and palmitic acid contents. However, the linoleic acid content had a significant positive correlation with the palmitic acid content and a significant negative correlation with the stearic acid content. It was concluded that oleic acid content in C. oleifera oil could affect linoleic acid and palmitic acid contents, and linoleic acid content in C. oleifera oil could affect stearic acid and palmitic acid contents. Correlation analysis is a measure of the correlation of two numerical type variables, and to calculate the magnitude of the correlation. Significance testing allows analysis of whether two variables have a significant linear correlation. The aim of this mathematical analysis was to analyze whether the four fatty acids affect each other and correlate with each other. According to the reviewer’s comment, we have made correction on line 241-242 in the revised manuscript and are marked in red.

Comment 26: 227 The term “bioactive components” is not precise. Fatty acids are also bioactive components, but reported in the section above. From my point of view the following components are minor compounds from the oil or polar compounds from the extraction meal.

Response:

Thanks for your kind suggestion and we apologize for the unclear language expression. According to the reviewer’s comment, we have made correction on lines 244, 245, 246, 257 in the revised manuscript and are marked in red.

Comment 27: 229 The relative peak area of the total ion chromatogram (TIC) of tentatively identified components of different samples were compared. No content was determined from my point of view. Please revise the sentence.

Response:

Thanks for your kind suggestion and we apologize for the unclear language expression. We changed this sentence on line 246-249 to “The relative contents of polar compounds were calculated by normalization of chromatographic peak area”. According to the reviewer’s comment, we have made correction on lines 246-249 in the revised manuscript and are marked in red.

Comment 28: 230 Vitamin E is a name for a group of substances. Please specify in detail, which tocopherol was meant.

Response:

Thank you for this valuable suggestion. Vitamin E, an umbrella term for tocopherols, tocotrienols and derivatives with physiological activities of tocopherols, is an important nutrient in maintaining various normal physiological activities of the human body. Vitamin E has four homologues in plants (α, β, γ, and δ tocopherol). Among them α- tocopherol has the strongest physiological activity, γ- tocopherol has the strongest antioxidant capacity [12]. Vitamin E detected in this study was α- tocopherol. According to the reviewer’s comment, we have made correction on line 246, 248, 253, 320, 359, 361, 363, 439 in the revised manuscript and are marked in red.

Reference

[12] Zou, Y.J. Geographical Patterns of Variations in Main Chemical Components of Oil-tea Camellia Seeds. Nanchang University, Nanchang, China, 2019. https://doi.org/10.27232/d.cnki.gnchu.2019.000886.

Comment 29: 230 You only mention stigmasterol, but not β-sitosterol. As β-sitosterol is the most important sterol in most edible oils, it would be of importance to mention any difference regarding camellia seed oil. Please discuss in more detail.

Response:

Thank you for your valuable question. We are so sorry to make the reviewers and other audiences confusing. The stigmasterol we mentioned in the manuscript was β-sitosterol. The substance we detected was β-sitosterol, because their structural formula was too similar, and we mistaken β-sitosterol for stigmasterol. The difference in relative contents of β-sitosterol between the 40 samples were not as great as that of squalene, and those with relative contents greater than 6% had BB, GZ, HN, and QZ4. According to the reviewer’s comment, we have made correction on line 246, 359 and line 251-253 in the revised manuscript and are marked in red.

Comment 30: 232-234 In the case that these effects are the same as in the introduction, this can be abbreviated. For new claims there should be a reference.

Response:

Thank you for this valuable suggestion. Surely, these effects of squalene are the same as in the introduction. And we supplemented the effects of vitamin E on lines 49-53 in the introduction: Vitamin E is capable of protecting T lymphocytes and erythrocytes, inhibiting free radical oxidation, and reducing the risk of myocardial and cerebral infarction [12]. Therefore, we changed this sentence on lines 248-249 to “because they all have good medicinal values”. According to the reviewer’s comment, we have made correction on lines 49-53 and lines 248-249 in the revised manuscript and are marked in red.

Reference

[12] Zou, Y.J. Geographical Patterns of Variations in Main Chemical Components of Oil-tea Camellia Seeds. Nanchang University, Nanchang, China, 2019. https://doi.org/10.27232/d.cnki.gnchu.2019.000886.

Comment 31: 240 Figure 3: Please specify in more detail, how you evaluated the GC-MS chromatogram. According to the procedure the oil was diluted, filtered and injected. However, the oven program of the GC run was stopped after reaching 270 °C. What happened with the triacylglycerols injected into the analysis system? In the diagram C of the TIC we see relative peak areas of up to 10 % for stigmasterol, 15 % for tocopherols (without silylation?), squalene up to more than 20 % and β-amyrin also up to more than 20 %. As these minor compounds are normaly detected at a very small level, there should be a lot of peaks disregarded. Please specify in more detail the used procedure.

Response:

Thank you for your valuable question. We are so sorry to make the reviewers and other audiences confusing. The procedure for GC-MS was as follows. Camellia spp. oil (0.1 g) was diluted with positive hexane to 5 mL and filtered through a membrane solution filter (0.22 μm) prior to GC-MS analysis. The GC conditions were as follows. The mixture was analyzed using an Agilent 7890B-7000B gas chromatograph equipped with an HP-5MS column (30 m × 0.25 mm × 0.25 μm) under the following temperature conditions: 60 ℃ for 1 min, followed by ramping of 6 ℃·min-1 to 270 ℃, and then maintenance at 270 ℃ for 2 min. The transfer line and ion resource temperatures were set to 250 ℃. The injected volume was 1 μL, splitless. The flow rate of pure helium (99.99%), the carrier gas, was 1.00 mL·min-1

The mass spectrometry conditions were as follows. The ion source was EI and the temperature 230 ℃. The quadrupole temperature was 150 ℃, ionization voltage was 70 eV, and the emission current was 34.60 μA. The multiplier voltage was 2000 V.

Data were obtained continuously in the full-scan mode in the mass range of 50-450 (m/z) [27]. On the HPMSD chemical workstation, compounds were tentatively identified using the NIST2005 MS and WILEY275 MS libraries, and their relative contents were calculated by normalization of chromatographic peak area [28].

We did detect a lot of substances in our GC-MS detection, and others included tetraethyl silicate; undecane; 2-Propenoic acid, 3-phenyl-, methyl ester; methyleugenol; dibutyl phthalate; hexadecanoic acid, ethyl ester; linoleic acid ethyl ester; (E)-9-octadecenoic acid ethyl ester; octadecanoic acid, ethyl ester; glycidyl palmitate; glycidyl oleate; bis(2-ethylhexyl) phthalate and so on. The triacylglycerols injected into the analysis system were turned into esters such as tetraethyl silicate, dibutyl phthalate, linoleic acid ethyl ester, as well as other substances (methyleugenol). Because we focused on the four substances (squalene, α- tocopherol, β-sitosterol, and β-amyrin), only the relative contents of the four substances were listed in the manuscript. According to the reviewer’s comment, we have made correction on line125-128 and line 256-257 in the revised manuscript and are marked in red.

References

[27] Ye, Z.C. The comparative study on chemical constituents and bioactivities of Camellia oleifera oils and cakes from different locations. Hainan University, Haikou, China, 2017.

[28] Adams, R. Identification of essential oil components by Gas Chromatography/Quadrupole Mass Spectroscopy. USA: Allured Publishing Corp, 2001.

Comment 32: Reference 35 is not significant for the statement in the sentence as it deals with the analysis of triacylglycerols using HPLC. Please refer to an adequate refence. In addition, the internet address lead to a completely differing article.

Response:

Thank you for this valuable suggestion and we are very sorry for our negligence regarding the citations. We have cited the correct reference [40] for the sentence on lines 567-569. According to the reviewer’s comment, we have made correction on lines 567-569 and the thirty-five reference in the revised manuscript and are marked in red.

References

[40] Wang, Y.P.; Fei, X.Q.; Yao, X.Y.; Wang, K.L.; Guo, S.H.; Ren, H.D. Principal component analysis and cluster analysis of fatty acids and triglycerides in oil - tea camellia seeds from different origins. China oils and fats. 2021, 46(09): 112-119. https://doi.org/10.19902/j.cnki.zgyz.1003-7969.200640.

Reviewer 2 Report

Please check the file

Author Response

Dear reviewer,

Thank you very much for forwarding the reviewers’ comments and giving us an opportunity to improve our manuscript (ID: 1818153). All suggestions have been carefully considered and the comments have been addressed point by point. The manuscript has been revised accordingly in red font. The responses to the reviewers’ comments are given below.

Reviewer #2:

Comment 1: Line 18: “Indexes” should have changed to “Indices”.

Response:

Thank you for this kind and valuable suggestion and we apologize for the unclear language expression. According to the reviewer’s comment, we have made correction on line 18 in the revised manuscript and are marked in red.

Comment 2: Line 43-44: “β- Amyrin” should have changed to “β- amyrin”.

Response:

Thank you for this kind and valuable suggestion and we apologize for the unclear language expression. According to the reviewer’s comment, we have made correction on line 46 in the revised manuscript and are marked in red.

Comment 3: Line 45: It is advisable to remove the model verb can and replace it with the present simple tense.

Response:

Thank you for the valuable suggestion. According to the reviewer’s comment, we have made correction on line 48 in the revised manuscript and are marked in red.

Comment 4: Line 48: If you please, end the sentence with industries. Please, consider mentioning reported phenolic compounds from Camellia oleifera spp. With an emphasis on their bioactivities. It is advisable to mention some of the analytical methods of reported fatty acids and phenolics.

Response:

Thank you for the valuable suggestion, and we apologize for the unclear language expression. In the materials methods block, references were cited for the majority of methods of reported fatty acids and bioactive compounds. In accordance with the Reviewer’s comment, we have made correction on line 55 in the revised manuscript and are marked in red. And we have added some points about the reported bioactive aspects of tocopherols from the genus Camellia on line 49-53 in the revised manuscript and are marked in red.

Comment 5: Line 77: “Camellianin A” should have changed to “camellianin A”.

Response:

Thank you for this kind and valuable suggestion and we apologize for the unclear language expression. According to the reviewer’s comment, we have made correction on line 86 in the revised manuscript and are marked in red.

Comment 6: Line 95: “n-hexane” should have changed to “n-hexane”. It is advisable to cite articles on the methods whenever possible.

Response:

Thank you for this kind and valuable suggestion and we apologize for the unclear language expression. We have changed n-hexane into n-hexane. And we have added references (such as reference 26) for the methods on line 106. According to the reviewer’s comment, we have made correction on lines 104, 118 in the revised manuscript and are marked in red.

References

[26] O'Fallon, J.V.; Busboom, J.R.; Nelson, M.L.; Gaskins, C.T. A direct method for fatty acid methyl ester synthesis: Application to wet meat tissues, oils, and feedstuffs. Journal of Animal Science. 2007, 85(6): 1511-1521. https://doi.org/10.2527/jas.2006-491.

Comment 7: Line 133: “Standard sample” should have changed to “Standard.”. Determination of total phenolic and total flavonoid content: Please clarify the sample you are working on

Response:

Thank you for your valuable question. We are so sorry to make the reviewers and other audiences confusing. The 0.3 g C. oleifera spp. degreasing powder was extracted by 4.5 mL 50% methanol extraction solution, then treated by ultrasonic wave (water bath 60 ℃, power 100 W, frequency 40 kHz) for 30 min, filter, repeat the above steps for 3 times, combine the filtrate, vacuum concentrate at 45 ℃, add equal volume of extraction solution, centrifuge at 1697×g for 15 min, suck the supernatant, and store at 20 ℃ for use [27]. According to the reviewer’s comment, we have made correction on line 145-149 and line 151 in the revised manuscript and are marked in red.

References

[27] Ye, Z.C. The comparative study on chemical constituents and bioactivities of Camellia oleifera oils and cakes from different locations. Hainan University, Haikou, China, 2017.

Comment 8: Line 140: “wave(water” should have changed to “wave (water”.

Response:

Thank you for this kind and valuable suggestion and we apologize for the unclear language expression. According to the reviewer’s comment, we have made correction on line 158 in the revised manuscript and are marked in red.

Comment 9: Line 165, 170: “FeSO4” should have changed to “FeSO4”.

Response:

Thank you for this kind and valuable suggestion and we apologize for the unclear language expression. According to the reviewer’s comment, we have made correction on line 184, 189 in the revised manuscript and are marked in red.

Comment 10: Line 198: “samples studied” should have changed to “studied samples”.

Response:

Thank you for this kind and valuable suggestion and we apologize for the unclear language expression. According to the reviewer’s comment, we have made correction on lines 220 in the revised manuscript and are marked in red.

Comment 11: Line 230: “Kinds” should have changed to “Varieties”.

Response:

Thank you for this kind and valuable suggestion and we apologize for the unclear language expression. According to the reviewer’s comment, we have made correction on line 246-249 in the revised manuscript and are marked in red.

Comment 12: Line 232-234: Citation is needed

Response:

Thank you for this valuable suggestion. These effects of squalene [11] are the same as in the introduction. And we supplemented the effects of vitamin E on lines 49-53 in the introduction: Vitamin E is capable of protecting T lymphocytes and erythrocytes, inhibiting free radical oxidation, and reducing the risk of myocardial and cerebral infarction [12]. Therefore, we changed this sentence on lines 248-249 to “because they all have good medicinal values”. According to the reviewer’s comment, we have made correction on lines 49-53 and lines 248-249 in the revised manuscript and are marked in red.

References

[11] Cao, Y.; Xie, Y.; Ren, H. Fatty acid composition and tocopherol, sitosterol, squalene components of Camellia reticulata oil. Journal of Consumer Protection and Food Safety. 2018, 13 (4), 403-406. https://doi.org/10.1007/s00003-018-1183-8.

[12] Zou, Y.J. Geographical Patterns of Variations in Main Chemical Components of Oil-tea Camellia Seeds. Nanchang University, Nanchang, China, 2019. https://doi.org/10.27232/d.cnki.gnchu.2019.000886.

Reviewer 3 Report

The current manuscript describes fatty acid composition and antioxidant properties of 40 Camellia species, yet some comments needed to be considered:

1. My major concern is the nomenclature in the title, abstract and whole manuscript, you used 40 Camellia species and not Camellia oleifera spp. Camellia oleifera is the name of a species not genus and you didn't even use varieties of Camellia oleifera, they are different species as C. vietnamensis, C. gigantocanzoa, C.chekiangoleosa,...etc.

2. In the plant material section, taxonomist details who made the identification is missing as well as voucher specimens codes and place

3.line 86, 87: is the temp used for extraction 60-90 or 70-90 and is there a reference for using this high temperature for 5 hrs.

4.line 108: what do you mean by positive hexane?

5.line 180: kindly specify whether the 46.87% is oil content v/w or w/w?

6.Figure 2B and 3B are not cited in the text 

7. Another concern is that you mentioned determination and identification of peaks by peak area which is incorrect. Peak area can be used for quantification and not identification, as mentioned in line 196, 230

8. I couldn't understand what you meant by standard product in Table S1 and Fig. S1. Kindly clarify

9.In table S2, total phenols are not a standard as well as total flavonoids, DPPH, FRAP, and ABTS. Instead, I suggest writing gallic acid, rutin, trolox,..etc

10. Table S3, kindly write the names of major constituents 1-15 instead of numbers for no confusion to the readers

Author Response

Dear reviewer,

Thank you very much for forwarding the reviewers’ comments and giving us an opportunity to improve our manuscript (ID: 1818153). All suggestions have been carefully considered and the comments have been addressed point by point. The manuscript has been revised accordingly in red font. The responses to the reviewers’ comments are given below.

Reviewer #3:

Comment 1: My major concern is the nomenclature in the title, abstract and whole manuscript, you used 40 Camellia species and not Camellia oleifera spp. Camellia oleifera is the name of a species not genus and you didn't even use varieties of Camellia oleifera, they are different species as C. vietnamensis, C. gigantocanzoa, C.chekiangoleosa,...etc.

Response:

Thank you for your valuable question. We are so sorry to make the reviewers and other audiences confusing. Surely, we used 40 Camellia species as  experimental materials. They are not varieties of Camellia oleifera and are different species (Liu et al., 2021; Dong et al. 2022). The Camellia oleifera spp. in the manuscript have changed into Camellia spp. According to the reviewer’s comment, we have made correction in the revised manuscript and are marked in red.

References

Liu, R.-C.; Xiao, Z.-Y.; Hashem, A.; Abd_Allah, E.F.; Xu, Y.-J.; Wu, Q.-S. Unraveling the Interaction between Arbuscular Mycorrhizal Fungi and Camellia Plants. Horticulturae. 2021, 7, 322. https://doi.org/10.3390/horticulturae7090322.

Dong, B.; Deng, Z.; Liu, W.; Rehman, F.; Yang, T.-J.; Huang, Y.F.; Gong, H.G. Development of expressed sequence tag simple sequence repeat (EST-SSR) markers and genetic resource analysis of tea oil plants (Camellia spp.). Conservation Genet Resour. 2022, 14, 41–45. https://doi.org/10.1007/s12686-021-01248-x.

Comment 2: In the plant material section, taxonomist details who made the identification is missing as well as voucher specimens codes and place.

Response:

Thank you for this kind and valuable suggestion. The seed samples were harvested in 2018, identified and provided by Prof. Kaibing Zhou. The samples were not collected in the curina and were taken locally and therefore had no voucher species codes. According to the reviewer’s comment, we have made correction on line 80-81 in the revised manuscript and are marked in red.

Comment 3: line 86, 87: is the temp used for extraction 60-90 or 70-90 and is there a reference for using this high temperature for 5 hrs.

Response:

Thank you for your valuable question. We are so sorry to make the reviewers and other audiences confusing. We have written the extraction temperature incorrectly as 70–90 ℃, which should be 50 ℃. The processed seeds were loaded into the filter paper package (Soxhlet extractor) with petroleum ether (boiling point: 60–90 ℃) for extraction at 50 ℃ for 5 h to obtain the oil and to obtain a deoiled powder [23-25]. According to the reviewer’s comment, we have made correction on line 95-97 in the revised manuscript and are marked in red.

References  

[23] Al Juhaimi, F.; Özcan, M.M.; Ghafoor, K.; Babiker, E.E.; Hussain, S. Comparison of cold-pressing and soxhlet extraction systems for bioactive compounds, antioxidant properties, polyphenols, fatty acids and tocopherols in eight nut oils. Journal of Food Science and Technology. 2018, 55, 3163–3173. https://doi.org/10.1007/s13197-018-3244-5.

[24] Özcan, M.M.; Al-Juhaimi, F.Y.; Ahmed, I.A.M.; Osman, M.A.; Gassem, M.A.  Effect of soxhlet and cold press extractions on the physico-chemical characteristics of roasted and non-roasted chia seed oils. Food Measure. 2019, 13, 648–655. https://doi.org/10.1007/s11694-018-9977-z.

[25] Desai, S.N.; Jadhav, A.J.; Holkar, C.R.; Pawar, B.G.; Pinjari, D.V. Extraction and microencapsulation of Buchanania lanzan Spreng seed oil. Chemical Papers. 2022, 76, 3521–3530. https://doi.org/10.1007/s11696-022-02116-0.

Comment 4: line 108: what do you mean by positive hexane?

Response:

Thank you for this valuable suggestion and we apologize for the unclear language expression. The "positive hexane" is n-hexane. According to the reviewer’s comment, we have made correction on line 118 in the revised manuscript and are marked in red.

Comment 5: line 180: kindly specify whether the 46.87% is oil content v/w or w/w?

Response:

Thank you for the valuable suggestion. The 46.87% is oil content w/w. In accordance with the reviewer’s comment, we have made correction on line 198 in the revised manuscript and are marked in red.

Comment 6: Figure 2B and 3B are not cited in the tex

Response:

Thank you for your valuable question. We are so sorry to make the reviewers and other audiences confusing. Figure 2B and 3B have changed into Figure 3B and 3D which have been cited in the manuscript. In accordance with the reviewer’s comment, we have made correction on line 334 and line 341-343 in the revised manuscript and are marked in red.

Figure 3. (A): The heatmap of fatty acids in Camellia spp., respectively; (B): The PCA analysis of fatty acids in Camellia spp., respectively; (C): The heatmap of active content in Camellia spp., respectively; (D): The PCA analysis of active content in Camellia spp., respectively.

Comment 7: Another concern is that you mentioned determination and identification of peaks by peak area which is incorrect. Peak area can be used for quantification and not identification, as mentioned in line 196, 230.

Response:

Thank you for your valuable question. We are so sorry to make the reviewers and other audiences confusing. Surely, Peak area can be used for quantification and not identification. We determined and identified peaks by their time of appearance. In accordance with the reviewer’s comment, we have made correction on line 217-218 and line 247 in the revised manuscript and are marked in red.

Comment 8: I couldn't understand what you meant by standard product in Table S1 and Fig. S1. Kindly clarify.

Response:

Thank you for your valuable question. We are so sorry to make the reviewers and other audiences confusing. Figure S1 and Table S1 were the standard sample fatty acid composition analysis chromatogram with its peak retention time. After obtaining the gas chromatograms of 40 samples to find the peak where the fatty acid in the sample was located following the chromatographic contrast of the standards, area normalization was used to express the area ratio and calculate the relative content of the fatty acids contained. I have marked the numbers in Figure S1, corresponding to Table S1. In accordance with the Reviewer’s comment, we have made correction in Figure S1 in the revised manuscript and are marked in red.

Comment 9: In table S2, total phenols are not a standard as well as total flavonoids, DPPH, FRAP, and ABTS. Instead, I suggest writing gallic acid, rutin, trolox,..etc.

Response:

Thank you for your valuable question. We are so sorry to make the reviewers and other audiences confusing. In accordance with the Reviewer’s comment, we have made correction in table S2 in the revised manuscript and are marked in red.

Comment 10: Table S3, kindly write the names of major constituents 1-15 instead of numbers for no confusion to the readers.

Response:

Thank you for your valuable question. We are so sorry to make the reviewers and other audiences confusing. In accordance with the Reviewer’s comment, we have made correction in Table S3 in the revised manuscript and are marked in red.

Round 2

Reviewer 3 Report

The authors have addressed all comments. The manuscript is now suitable for publication.